# Scalable Autoregressive 3D Molecule Generation

## Abstract

Generative models of 3D molecular structure play a rapidly growing role in the design and simulation of molecules. Diffusion models currently dominate the space of 3D molecule generation, while autoregressive models have trailed behind. In this work, we present Quetzal, a simple but scalable autoregressive model that builds molecules atom-by-atom in 3D. Treating each molecule as an ordered sequence of atoms, Quetzal combines a causal transformer that predicts the next atom's discrete type with a smaller Diffusion MLP that models the continuous next-position distribution. Compared to existing autoregressive baselines, Quetzal achieves substantial improvements in generation quality and is competitive with the performance of state-of-the-art diffusion models. In addition, by reducing the number of expensive forward passes through a dense transformer, Quetzal enables significantly faster generation speed, as well as exact divergence-based likelihood computation. Finally, without any architectural changes, Quetzal natively handles variable-size tasks like hydrogen decoration and scaffold completion. We hope that our work motivates a perspective on scalability and generality for generative modelling of 3D molecules. Code is available at https://anonymous.4open.science/r/quetzal-5BD3.

## 1 Introduction

Generative models of 3D molecular structure are accelerating the design and simulation of molecules, with applications across chemistry, biology, and materials science (Abramson et al., 2024; Watson et al., 2023; Zeni et al., 2025). Diffusion-based approaches are the prevailing standard (Hoogeboom et al., 2022; Song et al., 2024b; Zhang et al., 2024; Joshi et al., 2025), but they typically operate on fixed-size input/output and are computationally intensive to sample from. In contrast, autoregressive models of 3D molecules have lagged behind in generation quality (Gebauer et al., 2019; Luo & Ji, 2022; Daigavane et al., 2023; Flam-Shepherd & Aspuru-Guzik, 2023; Gao et al., 2024). However, autoregressive models offer several compelling advantages: they support arbitrary-size generation, enable exact likelihood computation, and offer potentially faster generation. Moreover, molecules are naturally tokenized into atoms, which aligns with the paradigm of autoregression.

This performance gap is often attributed to the assumption that diffusion models are suited for continuous spatial data, whereas autoregressive models are designed for discrete domains like text. Indeed, prior autoregressive methods for 3D structure typically *discretize* coordinates into 3D grids or tokenized .xyz files, discarding important information about spatial continuity. However, recent work by Li et al. (2024a) has challenged this assumption by introducing a Diffusion Loss, which jointly trains a lightweight, per-token diffusion model with an autoregressive transformer. This hybrid architecture enables autoregressive generation of continuous-valued tokens while retaining the scalability of transformers.

In this work, we adapt per-token diffusion for 3D molecule generation. We propose Quetzal, a simple yet scalable autoregressive model that generates molecules atom-by-atom, predicting each atom's discrete type and continuous 3D position. Quetzal combines a causal transformer with a smaller Diffusion MLP to model the position of the next atom, conditioned on the current prefix structure. This simple design enables Quetzal to scale, achieving generation quality that surpasses all autoregressive baselines and competes with state-of-the-art diffusion models, while also

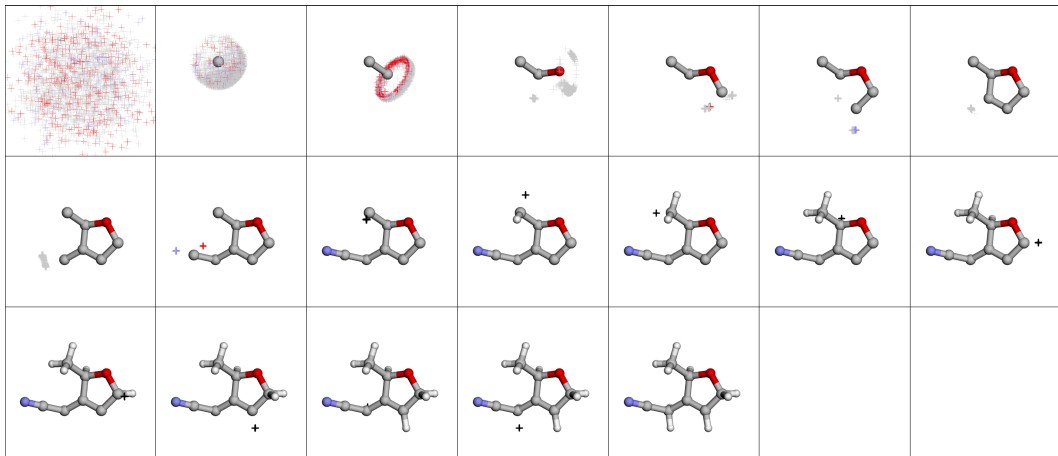

Figure 1: QUETZAL generates 3D molecules by iteratively predicting the next atom's discrete type and continuous position. Cross marks indicate the distribution of the next atom's type and position.

significantly improving generation speed. Furthermore, without additional training, QUETZAL automatically performs flexible generation tasks such as hydrogen decoration and scaffold completion, which are cumbersome to implement with fixed-size diffusion models. By revisiting autoregression through the lens of modern scaling but with a continuous spatial inductive bias, QUETZAL repositions autoregressive models as a competitive and versatile approach for 3D molecule generation.

## 2 RELATED WORK

**3D molecular generative models.** Generative models of 3D molecules have been proposed using normalizing flows (Garcia Satorras et al., 2021), diffusion models (Hoogeboom et al., 2022), flow matching (Song et al., 2024b), Bayesian flow networks (Song et al., 2024a), and latent diffusion (Xu et al., 2023; Joshi et al., 2025). Further works have enhanced generation capability by leveraging representation conditioning (Li et al., 2024b) or optimal transport (Hong et al., 2024). These approaches are often designed around equivariant architectures and typically model molecules as unordered point clouds. Other representations such as voxel grids (O Pinheiro et al., 2024) and neural fields (Kirchmeyer et al., 2025) move beyond fixed-size generation, but lose the sparse representation of point clouds.

**Autoregressive 3D molecular generative models.** Autoregressive models such as G-SchNet (Gebauer et al., 2019) and Symphony (Daigavane et al., 2023) discretize 3D coordinates and predict relative positions using equivariant architectures. Other models (Luo & Ji, 2022; Liu et al., 2022) predict continuous quantities using normalizing flows or mixture models, but remain tied to reference frames. Another line of work simply casts 3D generative modelling as discrete language modelling of raw `.xyz` files (Flam-Shepherd & Aspuru-Guzik, 2023; Gruver et al., 2024; Zholus et al., 2024; Gan et al., 2025) or using custom tokenized representations (Wang et al., 2025a; Gao et al., 2024). Recently, UniGenX applies a similar autoregressive diffusion approach across molecular, crystals, and proteins (Zhang et al., 2025).

**Autoregression over continuous-valued tokens.** The most directly related work to ours is masked autoregression (MAR) (Li et al., 2024a), which introduces a Diffusion Loss for continuous-valued per-token generation in tandem with an autoregressive transformer backbone. Other approaches such as TimeGrad (Rasul et al., 2021) and Diffusion Forcing (Chen et al., 2024) apply similar prefix-conditional, per-token diffusion models for generating continuous-valued sequences. Instead of per-token diffusion, Jetformer predicts per-token Gaussian mixture parameters (Tschannen et al., 2024), and in this way trains a normalizing flow that understands text and images in data space. Trans-dimensional jump diffusion enables a diffusion model to add new dimensions during generation, which resembles autoregression (Campbell et al., 2024).

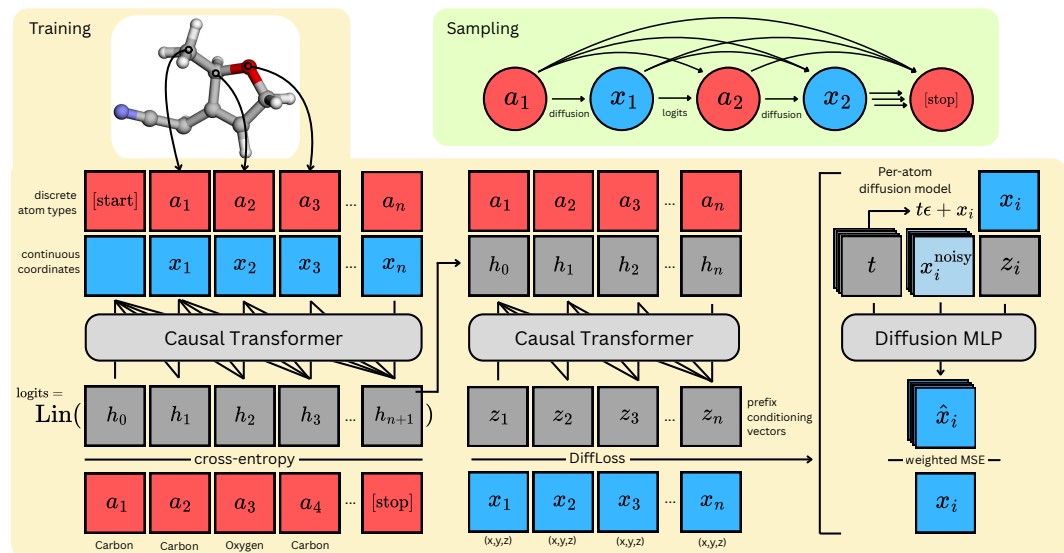

Figure 2: Architecture of QUETZAL during training and sampling. The first transformer stack causally processes each prefix of an input structure to predict logits of the next atom type. The second transformer stack incorporates information of the next atom type and causally produces conditioning vectors for the next 3D position. The prefix-conditional, per-token Diffusion MLP is trained jointly with the full transformer using multiple timesteps. Simultaneously batching across length and time provides dense supervision signal. Sampling iteratively predicts atom type and 3D position until [stop] is predicted.

## 3 ARCHITECTURE

Consider an $n$-atom 3D molecule $\mathcal{M} = (\boldsymbol{a}, \boldsymbol{x})$ as an *ordered* sequence of atom types $\boldsymbol{a} = (a_1, ..., a_n) \in \mathbb{N}^n$ and coordinates $\boldsymbol{x} = (x_i, y_i, z_i)_{i=1}^n = (\boldsymbol{x}_1, ..., \boldsymbol{x}_n) \in \mathbb{R}^{n \times 3}$. This atom ordering is taken directly from the original .xyz file. We refer to atoms and tokens interchangeably. Denote $\boldsymbol{x}_{:i} = (\boldsymbol{x}_1, ..., \boldsymbol{x}_i)$ as the *prefix* of $\boldsymbol{x}$ at sequence index $i$, which contains the current token and all previous tokens. We refer to $(\boldsymbol{a}_{:i}, \boldsymbol{x}_{:i})$ as the *prefix structure*. We index from 1, so $(\boldsymbol{a}_{:0}, \boldsymbol{x}_{:0})$ is just a dummy start token.

We define an autoregressive model of 3D molecules (Figure 1) which iteratively predicts the next atom's type and position $(a_{i+1}, \boldsymbol{x}_{i+1})$ given a prefix structure $(\boldsymbol{a}_{:i}, \boldsymbol{x}_{:i})$,

$$p(\mathcal{M}) = \prod_{i=0} p(a_{i+1}, \boldsymbol{x}_{i+1} | \boldsymbol{a}_{:i}, \boldsymbol{x}_{:i}) = \prod_{i=0} p_{\text{type}}(a_{i+1} | \boldsymbol{a}_{:i}, \boldsymbol{x}_{:i}) p_{\text{pos}}(\boldsymbol{x}_{i+1} | \boldsymbol{a}_{:i+1}, \boldsymbol{x}_{:i}). \quad (1)$$

The model alternates between predicting $a_i$ and $\boldsymbol{x}_i$ (i.e. sampling from $p_{\text{type}}$ and $p_{\text{pos}}$) until a [stop] token is sampled from $p_{\text{type}}$, or until a maximum number of atoms is reached. The generative model is fully specified when we specify these two conditional distributions. We now introduce each component of QUETZAL in order.

**Prefix embedding.** The next-type distribution $p_{\text{type}}(a_{i+1} | \boldsymbol{a}_{:i}, \boldsymbol{x}_{:i})$ is a categorical distribution, which is straightforward to model using a standard GPT. First, we embed the prefix structure $(\boldsymbol{a}_{:i}, \boldsymbol{x}_{:i})$ using embeddings for the atom types, linear layers and Fourier encodings (Tancik et al., 2020) for the coordinates, and learned positional encodings for sequence ordering. The combined embeddings are then passed through a causal transformer (Vaswani, 2017) to obtain prefix embeddings:

$$\boldsymbol{h}_i = \text{Transformer} \left( \text{Emb}(\boldsymbol{a}_{:i}) + \text{Lin}(\boldsymbol{x}_{:i}) + \text{Lin}(\text{Fourier}(\boldsymbol{x}_{:i})) + \text{PosEmb}_{:i} \right). \quad (2)$$

Because attention is causally masked, the causal transformer produces prefix embeddings $\boldsymbol{h}_i$ for all prefixes in the sequence in one forward pass (Figure 2).

**Next-type prediction.** Each prefix embedding $\boldsymbol{h}_i$ is passed to a linear layer with no bias which predicts logits of the next atom type,

$$p(a_{i+1} | \boldsymbol{a}_{:i}, \boldsymbol{x}_{:i}) = p(a_{i+1} | \boldsymbol{h}_i) = \text{softmax}(\text{Lin}(\boldsymbol{h}_i)), \quad (3)$$

which are supervised by cross-entropy loss against the ground truth next atom types.

**Prefix conditioning for diffusion.** We use the Diffusion Loss proposed by Li et al. (2024a) to model the continuous next-position distribution $p_{\text{pos}}(\boldsymbol{x}_{i+1}|\boldsymbol{a}_{:i+1}, \boldsymbol{x}_{:i})$. In other words, we model $p_{\text{pos}}$ as a single-atom-position diffusion model conditioned on a prefix structure $(\boldsymbol{a}_{:i}, \boldsymbol{x}_{:i})$ and next atom type $\boldsymbol{a}_{i+1}$. We first combine the prefix embeddings $\boldsymbol{h}_i$ with the next atom type $\boldsymbol{a}_{i+1}$ and pass through a second transformer to obtain $\boldsymbol{z}_i$, a conditioning vector which encodes the next-position distribution:

$$p_{\text{pos}}(\boldsymbol{x}_{i+1}|\boldsymbol{a}_{:i+1}, \boldsymbol{x}_{:i}) = p(\boldsymbol{x}_{i+1}|\boldsymbol{z}_{i+1}), \text{ where } \boldsymbol{z}_{i+1} = \text{Transformer}(\text{Emb}(\boldsymbol{a}_{:i+1}) + \boldsymbol{h}_{:i}). \quad (4)$$

This structure forces the model to commit to a discrete atom type before predicting its continuous position, which we find is beneficial for learning. Before defining $p(\boldsymbol{x}_{i+1}|\boldsymbol{z}_{i+1})$, we first introduce diffusion models, closely following the framework of Karras et al. (2022).

**Diffusion models** learn to sample from a continuous distribution defined by a dataset $p_{\text{data}}(\boldsymbol{x})$. Data is corrupted by adding Gaussian noise that grows as time $t$ increases: $\boldsymbol{x}_t = \boldsymbol{x} + t\boldsymbol{\varepsilon}$, where $\boldsymbol{\varepsilon} \sim \mathcal{N}(0, \boldsymbol{I})$. This corruption process defines a time-dependent probability density $p_t(\boldsymbol{x}) = p_{\text{data}}(\boldsymbol{x}) \otimes \mathcal{N}(\boldsymbol{0}, t^2\boldsymbol{I})$. At small times, $p_0$ approximates the data distribution, whereas for large time $T$, $p_T$ is well approximated as a large Gaussian $\mathcal{N}(\boldsymbol{0}, T^2\boldsymbol{I})$, which can be sampled without knowing $p_{\text{data}}$. Then, samples $\boldsymbol{x}_t \sim p_t(\boldsymbol{x}_t)$ can be generated by drawing samples $\boldsymbol{x}_T \sim p_T(\boldsymbol{x}_T)$ and evolving them backwards from time $T$ to $t$ under the probability flow ODE (Song et al., 2020b; Karras et al., 2022),

$$d\boldsymbol{x} = -t\nabla_{\boldsymbol{x}} \log p_t(\boldsymbol{x})dt, \quad (5)$$

which can be done as long as we know the time-dependent score function $\nabla_{\boldsymbol{x}} \log p_t(\boldsymbol{x})$. This score function is learned by a neural network $\boldsymbol{s}_\theta(t, \boldsymbol{x})$ with parameters $\theta$ by minimizing the denoising score matching objective (Vincent, 2011) for every $t$:

$$\mathbb{E}_{\boldsymbol{x} \sim p_{\text{data}}, \boldsymbol{\varepsilon} \sim \mathcal{N}(0, I)} ||\boldsymbol{s}_\theta(t, \boldsymbol{x} + t\boldsymbol{\varepsilon}) + \frac{\boldsymbol{\varepsilon}}{t}||^2. \quad (6)$$

By Tweedie's formula (Efron, 2011),

$$D(t, \boldsymbol{y}) = \boldsymbol{y} + t^2\nabla_{\boldsymbol{y}} \log p_t(\boldsymbol{y}), \quad (7)$$

this objective can be rewritten as learning an optimal denoiser $D_\theta(t, \boldsymbol{x}^{\text{noisy}})$ that aims to predict the original data $\boldsymbol{x}$ from the corrupted data $\boldsymbol{x}^{\text{noisy}} = \boldsymbol{x} + t\boldsymbol{\varepsilon}$,

$$\mathbb{E}_{\boldsymbol{x} \sim p_{\text{data}}, \boldsymbol{\varepsilon} \sim \mathcal{N}(0, I)} ||D_\theta(t, \boldsymbol{x} + t\boldsymbol{\varepsilon}) - \boldsymbol{x}||^2. \quad (8)$$

A diffusion model is readily extended to conditional distributions by simply providing a conditioning vector $\boldsymbol{z}$ as an extra input to the network.

**Per-atom diffusion.** Let $j = i+1$. We define the next-position distribution as a conditional diffusion model whose target distribution is $p(\boldsymbol{x}_{i+1}|\boldsymbol{z}_{i+1}) = p(\boldsymbol{x}_j|\boldsymbol{z}_j)$, giving the following objective for learning the next-position distribution:

$$\mathbb{E}_{\boldsymbol{x}_j \sim p_{\text{pos}}, \boldsymbol{\varepsilon} \sim \mathcal{N}(0, I)} ||D_\theta(t, \boldsymbol{x}_j + t\boldsymbol{\varepsilon}, \boldsymbol{z}_j) - \boldsymbol{x}_j||^2, \quad (9)$$

which is a restatement of DiffLoss (Li et al., 2024a). We predict $\hat{\boldsymbol{x}}_j = D_\theta(t, \boldsymbol{x}_j^{\text{noisy}}, \boldsymbol{z}_j)$ using a Diffusion MLP (DiffMLP) with adaptive layer normalization (Perez et al., 2017; Peebles & Xie, 2022), zero-initialization (Peebles & Xie, 2022), and residual connections (He et al., 2015):

$$\hat{\boldsymbol{x}}_j = \text{DiffMLP}\left(\text{Fourier}(t) + \text{Lin}(\boldsymbol{x}_j^{\text{noisy}}) + \text{Lin}(\text{Fourier}(\boldsymbol{x}_j^{\text{noisy}})) + \text{Lin}(\boldsymbol{z}_j)\right), \quad (10)$$

with Fourier encodings featurizing the low-dimensional inputs $\boldsymbol{x}_j^{\text{noisy}}$ and $t$. During training, once the conditioning vector $\boldsymbol{z}_j$ has been constructed, we independently sample 4 timesteps $t$ to expand the batch size used for training the DiffMLP. Once the denoiser is trained, one can access the score through Equation (7), and can sample $p_{\text{pos}}$ by drawing random noise and integrating Equation (5) from time $t = T$ to $t = 0$ using $N_{\text{diff}}$ discretized timesteps. $N_{\text{diff}}$ is significantly reduced by using the efficient sampling schedule proposed by Karras et al. (2022). See Appendix A for further details on sampling, preconditioning the neural network, and timestep-weighting during training.

**Combined loss.** We sum the cross-entropy and diffusion losses of all atoms together with no weighting. The entire network, consisting of the two causal transformer stacks and the DiffMLP, is trained

end-to-end. In this way, QUETZAL provides dense supervision on every atom type and position, batched across both sequence length and timesteps.

**Hybrid architecture.** QUETZAL separates the concerns of generative modelling into modelling quadratic-scaling atom *interdependence* with transformers, and modelling *individual* next-position distributions using a DiffMLP. This separation is analogous to how AlphaFold 3 uses a cubic-scaling trunk with a quadratic-scaling diffusion transformer (Abramson et al., 2024).

**Symmetries.** Molecules have translation, rotation, and permutation symmetries. To avoid overfitting to particular orientations, during training we apply simple data augmentation with random rotations and random translations of up to 3Å from the center-of-mass (Wang et al., 2024; Abramson et al., 2024; Tan et al., 2025a). We treat molecules as ordered sequences of atoms, and we inherit the ordering of atoms as listed in the `.xyz` file, similar to recent work (Yan et al., 2023; Vonessen et al., 2025). While not unique, this ordering is likely to have originated from how the original 3D structure was initialized from SMILES or drawn by hand, which importantly provides a *local ordering* of atoms. In Appendix B.3, we show that QUETZAL only relies on accessing local orderings, and not on specific dataset quirks.

**Inductive biases and scalability.** QUETZAL assigns greater priority to continuity rather than symmetry. Whereas previous autoregressive models discretize 3D space, QUETZAL's per-atom diffusion leverages the continuity of 3D space. The symmetries of 3D space are handled using data augmentation, rather than architectural equivariance, which relies on expensive tensor products or message-passing. This choice leaves QUETZAL free to use standard transformers and MLPs, which have scalable hardware implementations such as FlashAttention (Dao, 2023) and optimized kernels by `torch.compile` (Ansel et al., 2024). QUETZAL also operates in data-space and does not require learning a separate VAE tokenizer (Liu et al., 2024). Thus, QUETZAL accepts any 3D structure as input and generates arbitrary-size output, which enables flexible use for downstream tasks such as hydrogen decoration and scaffold completion.

**Fast generation.** Sampling from QUETZAL costs $n$ transformer calls (one per atom) and $nN_{\text{diff}}$ DiffMLP calls, each on an input of size 3. In contrast, all-atom diffusion spends $N_{\text{diff}}$ transformer or message-passing calls, each on an input of size $3n$. This architectural distinction enables significantly faster sampling for QUETZAL, especially on small molecules.

## 3.1 EXACT LIKELIHOOD ESTIMATION

In diffusion models, the change-of-variables formula (Chen et al., 2018) can be used to compute the log-likelihood $\log p_0(x_0)$ for a given data point $x_0$:

$$\log p_0(x_0) = \log p_T(x_T) + \int_0^T \nabla \cdot \boldsymbol{s}_\theta(t, x_t) \, \mathrm{d}t \tag{11}$$

Computing $\nabla \cdot \boldsymbol{s}_\theta(t, x_t)$, which is the divergence (trace-Jacobian) of the score function, requires computing $d$ vector-Jacobian products, where $d$ is the dimensionality of the data. In pure diffusion models, this is expensive because $d$ is large (e.g. $d = 3n \approx 132$ for GEOM with an average of 44 atoms). Therefore, most approaches resort to estimating log-likelihood via the ELBO or by approximating the divergence term using the Hutchinson trace estimator (Hutchinson, 1989). However, since our DiffMLP operates on 3-dimensional data, it is tractable to compute exact log-likelihood for each atom position by explicitly computing the full $3 \times 3$ Jacobian.

## 4 EXPERIMENTS

### 4.1 MOLECULAR GENERATION

We train QUETZAL on unconditional 3D molecular generation from the QM9 (Ramakrishnan et al., 2014) and GEOM-DRUGS (Axelrod & Gomez-Bombarelli, 2022) datasets (abbreviated as GEOM). We follow the train/val/test splits of Hoogeboom et al. (2022), as well as their evaluation protocol of generating 10,000 molecules and assessing atom stability, molecule stability, and validity and uniqueness via bond lookup tables. We also assess validity and uniqueness via xyz2mol as introduced by Daigavane et al. (2023). Finally, we assess negative log-likelihood (NLL) of the test set according to each model. We implement QUETZAL using a (6+6)-layer transformer (12 attention

heads, hidden size 768, 86M parameters) and a 6-layer DiffMLP (hidden size 1536, 79M parameters), resulting in 165M total parameters. More details on training, metrics, evaluation, generated samples, and ablation studies are in Appendix B.

Table 1: Sample quality of unconditionally generated molecules from QM9 by validity and uniqueness when generating 10,000 examples. Results for QUETZAL are means and standard deviations across 3 evaluation runs. QUETZAL uses $N_{\text{diff}} = 60$. Results on xyz2mol taken from (Gao et al., 2024), other results from respective works or *from our own evaluation. Higher is better, except for NLL. Best metrics overall are in **bold**, and best metrics out of autoregressive models are underlined.

| | atom stable | mol stable | lookup valid | lookup valid×uniq | xyz2mol valid | xyz2mol valid×uniq | NLL (↓) |
|---|---|---|---|---|---|---|---|
| QM9 | 99.36 | 95.30 | 97.67 | 97.63 | 99.99 | 99.90 | |
| EDM | 98.7 | 82.0 | 91.9 | 90.7 | 86.7 | 86.0 | −110.70 |
| GeoLDM | 98.9 | 89.4 | 93.8 | 92.7 | 91.3 | 90.3 | − |
| GeoBFN | **99.3** | **93.3** | 96.9 | 92.4 | − | − | − |
| SymDiff | 98.9 | 89.7 | 96.4 | **94.1** | 92.8* | 91.4* | **−133.79** |
| G-SchNet | 95.7 | 68.1 | 85.5 | 80.3 | 75.0 | 72.5 | − |
| Symphony | 90.8 | 43.9 | 68.1 | 66.5 | 83.5 | 81.8 | − |
| Mol-StrucTok | 98.5 | 88.3 | 98.0 | 83.4 | 96.7 | 82.5 | − |
| QUETZAL | 98.7 ±0.0 | 90.4 ±0.4 | 95.7 ±0.2 | 90.2 ±0.2 | **98.6** ±0.1 | **94.0** ±0.3 | −97.03 |

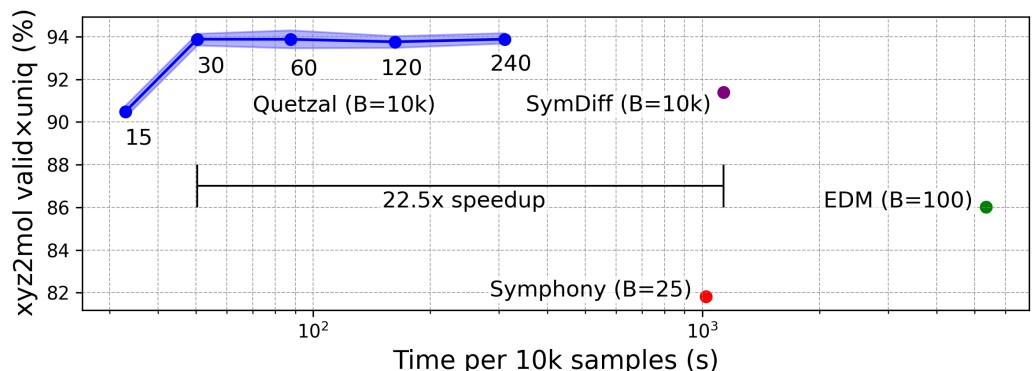

Figure 3: xyz2mol validity×uniqueness of 10k samples as a function of generation speed for QM9. $B$ is the batch size used for generation. We show the largest batch size that fits on a single A100 40 GB GPU. For QUETZAL, the text annotation is the number of diffusion steps used per atom. Error bars show min/max over 5 evaluations. See Appendix Figure 7 for generation speed vs batch size.

**Baselines**. We compare to equivariant diffusion models EDM (Hoogeboom et al., 2022), GeoLDM (Xu et al., 2023), GeoBFN (Song et al., 2024a), and SymDiff (Zhang et al., 2024). EDM, GeoLDM, and GeoBFN use equivariant graph neural networks (EGNNs) (Satorras et al., 2021), whereas SymDiff uses a permutation-equivariant diffusion transformer that scalably incorporates rotation equivariance via stochastic symmetrization. We also compare to autoregressive models G-SchNet (Gebauer et al., 2019) and Symphony (Daigavane et al., 2023), which rely on equivariant, relative predictions of the next atom position, as well as Mol-StrucTok (Gao et al., 2024), which tokenizes 3D structures into an SE(3)-invariant line notation for language modeling. All autoregressive baselines discretize 3D space.

## 4.2 QM9 GENERATION RESULTS

**Validity and uniqueness.** QUETZAL achieves strong sample quality on QM9, outperforming prior autoregressive methods in both xyz2mol and lookup table metrics (Table 1), and surpassing pure

Table 2: Sample quality of unconditionally generated molecules from GEOM by validity and uniqueness. QUETZAL uses $N_{\text{diff}} = 120$ for generation and $N_{\text{diff}} = 60$ for NLL. *We assume uniqueness is 100%.

| | atom stable | lookup valid | lookup valid×uniq | NLL ($\downarrow$) |
|---|---|---|---|---|
| GEOM | 86.5 | 99.9 | 69.5 | |
| EDM | 81.3 | 92.6 | 92.6* | $-137.1$ |
| GeoLDM | 84.4 | 99.3 | 99.3* | $-$ |
| GeoBFN | 86.2 | 91.7 | 91.7* | $-$ |
| GCDM | 89.0 | 95.5 | 95.5* | $-234.3$ |
| SymDiff | 86.2 | 99.3 | 99.3* | $-301.2$ |
| QUETZAL | $86.7_{\pm 0.0}$ | $95.6_{\pm 0.1}$ | $95.3_{\pm 0.2}$ | $-313.6$ |

diffusion models in xyz2mol metrics. QUETZAL exhibits signs of overfitting as evidenced by high validity but reduced validity×uniqueness. QUETZAL also obtains a poor estimate of test-set log-likelihood, despite generating high-quality samples. These observations may stem from QUETZAL overfitting to the fixed atom orderings seen during training. Ablations in Appendix B.2 also show that scaling model size consistently improves generation metrics, and that local atom orderings and rototranslation data augmentations are crucial to performance.

**Generation efficiency.** QUETZAL generates molecules significantly faster than all baselines, despite having the most parameters. Figure 3 shows the tradeoff between sample quality and generation time on a single A100 40GB GPU, where each model is run with the largest batch size that fits in memory. At $N_{\text{diff}} = 30$, QUETZAL achieves a 22.5× speedup over SymDiff while obtaining better xyz2mol validity×uniqueness. Although recent samplers (Song et al., 2020a; Lu et al., 2022; Karras et al., 2022) can reduce inference steps for diffusion models, matching QUETZAL's speed would require reducing SymDiff's 1000 steps to fewer than 44 — while preserving quality. Importantly, QUETZAL could also benefit from such sampler improvements. In Appendix Figure 7, we show that generation throughput also scales well with batch size.

The speedup can be attributed to several factors: (1) Pure diffusion models spend $N_{\text{diff}}$ calls to a dense $(3n \rightarrow 3n)$ transformer, whereas QUETZAL only calls the transformer once per new atom and spends $nN_{\text{diff}}$ calls to a small $(3 \rightarrow 3)$ MLP. (2) The number of diffusion steps is largely reduced by using the Heun sampler and geometrically-spaced timesteps proposed by Karras et al. (2022). (3) Forward passes are cheaper because the optimized performance of FlashAttention (Dao, 2023) is much faster and uses much less memory than expensive message-passing steps of EGNN (Satorras et al., 2021) or tensor products for Symphony, which also enables larger batch sizes.

### 4.3 GEOM GENERATION

QUETZAL is, to our knowledge, the first autoregressive model demonstrated on the large and diverse GEOM dataset. We compare to diffusion-based baselines including GCDM (Morehead & Cheng, 2024). QUETZAL again achieves generation quality approaching that of diffusion models (Table 2), but with much faster sampling: QUETZAL ($N_{\text{diff}} = 120$) requires 11.9 minutes for 10k samples, whereas EDM requires 1,533 minutes (128× speedup) and GCDM requires 683 minutes (57× speedup). Interestingly, QUETZAL achieves state-of-the-art NLL on GEOM, despite underperforming on QM9. We hypothesize that this is due to random splitting of GEOM: The dataset includes up to 30 conformers per molecule, so most molecules in the test set have conformers that are seen in the training set. This also explains why lookup validity×uniqueness of the original training set is low. We show uncurated samples in Appendix Figure 10.

### 4.4 HYDROGEN DECORATION

Because QUETZAL generates atoms sequentially, with hydrogens typically last, it can be applied with no additional training to decorate 3D structures with missing hydrogens. This task is useful for adding hydrogens to 3D structures from X-ray crystallography, which often lack resolved hy-

Table 3: Method performance on adding hydrogens onto bare molecules from the test set of QM9. All results are from our own evaluation. *Checkpoint appears to be undertrained, see Appendix B.4.

| Method | Correct Num H | % < RMSD Å 0.5 | 0.1 | 0.05 |
|---|---|---|---|---|
| Olex2 | 62.7 | 57.1 | 7.8 | 0.1 |
| OpenBabel+Hydride | 88.4 | 79.0 | 42.9 | 12.7 |
| Symphony* | 46.9 | 43.6 | 34.9 | 23.8 |
| QUETZAL | **99.8** | **99.5** | **94.1** | **90.4** |

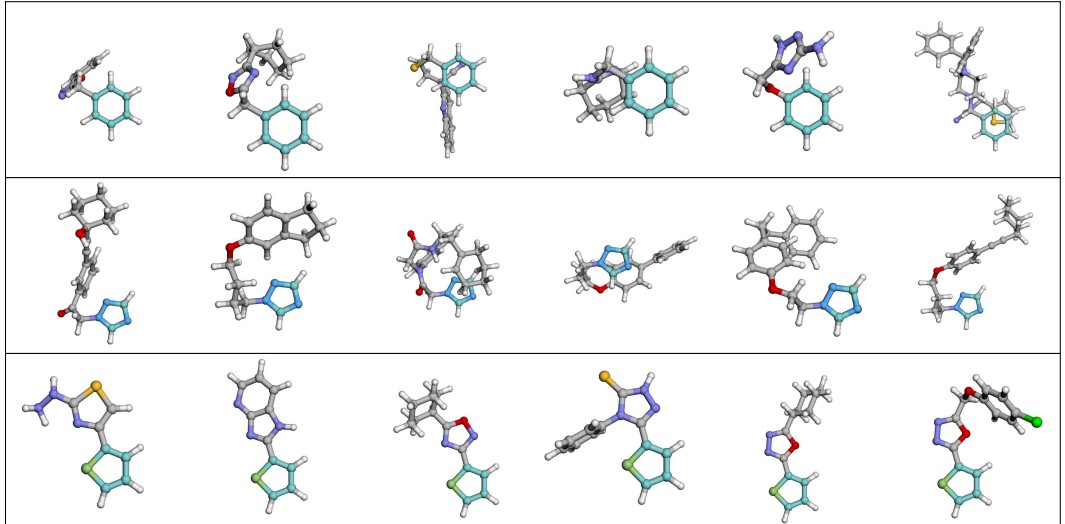

Figure 4: Selected examples of scaffold completion for benzene, 1,2,4-triazole, and thiophene. Generation uses $N_{\text{diff}} = 120$.

drogens due to low electron density (Müller, 2009). Usually, this task is performed with specialized cheminformatics software. It is unclear how to apply pure diffusion models to the task of hydrogen decoration, as they require specifying a fixed size. Therefore, we compare to tools such as the crystallography toolbox Olex2 (Dolomanov et al., 2009), and OpenBabel + Hydride (O'Boyle et al., 2011; Kunzmann et al., 2022), which first infers bonds then adds hydrogens in 3D. We also compare to Symphony, an autoregressive model trained on QM9. We test this task by stripping all hydrogens from the test set of QM9, and evaluate the accuracy in adding hydrogens back.

As metrics, we check whether each method adds the correct number of hydrogens. If the number of hydrogens is correct, we calculate the root-mean-squared deviation (RMSD) for just the hydrogen atoms, and check whether it satisfies thresholds of 0.5, 0.1, and 0.05 Å. Because hydrogens can be added in any order, we first assign permutations between predicted and ground truth by solving a linear assignment problem (Hungarian algorithm) on Euclidean distances. Results are in Table 3.

QUETZAL predicts hydrogens with high accuracy. However, QUETZAL's hydrogen predictions are sensitive to atom ordering. QUETZAL is able to solve this task because in QM9, hydrogens appear last in the `.xyz` files. If the bare molecule without hydrogens is reordered, QUETZAL's performance degrades, as the prefix becomes out-of-distribution. However, this degradation could be overcome by canonicalizing prefixes with a local atom order. Additional results are in Appendix B.4.

### 4.5 SCAFFOLD COMPLETION

Scaffold completion is naturally suited to autoregressive generation, since the prefix structure is held fixed. We demonstrate completions for benzene, 1,2,4-triazole, and thiophene scaffolds in Figure 4. These are qualitative results; we defer quantitative evaluation and comparison (Xie et al., 2024) to future work. We note that, like hydrogen decoration, scaffold completion is sensitive to the initial

scaffold configuration in terms of its atom ordering, center of mass, and orientation. However, this sensitivity can be useful to steer how the scaffold is completed.

## 5 DISCUSSION

Our architecture is simple: it does not require a separate tokenizer or autoencoder, does not model bonds explicitly, does not predict a focal atom or relative coordinates, and does not architecturally consider permutation, translation, or rotation symmetries. Instead, we rely on a standard transformer backbone equipped with Diffusion Loss, a simple method for autoregressive generation of per-token continuous coordinates (Li et al., 2024a). In doing so, we create a model that is simple to implement, trainable at scale, and fast to sample from.

Despite these advantages, one limitation of the model is its sensitivity to generation order. We show in Table 5 that QUETZAL only requires training on *local* orders, whereas training on more nonlocal orders reduces performance. These results reveal that order contains information, echoing trends seen in the text diffusion literature (Kim et al., 2025). Methods for inferring atom generation order during (Wang et al., 2025b) or after (Kim et al., 2025) training may overcome these challenges. Annealing or curriculum strategies may also help improve generation order robustness (Pannatier et al., 2024; Yu et al., 2025).

A separate limitation is that QUETZAL is not permutation-invariant, which degrades its performance on prefix-completion tasks, but not on unconditional generation. This is a consequence of using learned positional encodings. Although removing positional encodings does not confer permutation symmetry, since causal attention encodes token ordering (Haviv et al., 2022; Kazemnejad et al., 2024; Zuo et al., 2025), it may still improve generalization. Adding relative positional encodings may retain partial permutation-invariance while boosting length-generalization (Su et al., 2021; Loshchilov et al., 2024). Encoding order may even be a useful inductive bias in large, linear biomolecules such as proteins (Lin & AlQuraishi, 2023; Geffner et al., 2025). QUETZAL could be made permutation-invariant by using a non-causal transformer, but training throughput may drop since it prevents batching-in-sequence. However, it may be possible to finetune a causally-pretrained network into an attention-mask-free autoregressive network (Charpentier & Samuel, 2024).

Future work can exploit the fact that autoregressive models accept *arbitrary-size prompts* and generate *arbitrary-size responses* to provide extremely flexible conditioning in the form of text (Gruver et al., 2024). Future versions of QUETZAL could piggyback on innovations in autoregressive LLMs, such as accelerating generation speed using a kv-cache (Pope et al., 2023), or enhancing expressivity by reasoning over many tokens with chain-of-thought (Wei et al., 2022; Hao et al., 2024). In particular, reasoning could reduce sensitivity to generation order by enabling QUETZAL to reorder the prefix atoms itself. Likewise, context-extension methods developed for LLMs are a promising direction for improving length generalization of QUETZAL, and might enable even larger context windows than current all-atom diffusion approaches allow (Ruoss et al., 2023; Chen et al., 2023; Peng et al., 2023; Ding et al., 2024). Finally, tractable exact likelihood computation enables importance sampling for Boltzmann generators (Noé et al., 2019; Klein et al., 2023; Klein & Noé, 2024; Tan et al., 2025a;b), and could unlock new strategies for finetuning models on reward functions.

### REPRODUCIBILITY STATEMENT

We release all source code at `https://anonymous.4open.science/r/quetzal-5BD3`, which includes the architecture, training pipeline, data preprocessing with train/val/test splits, evaluation metrics, and Jupyter notebooks for reproducing figures. When deanonymized, we will also release trained checkpoints, preprocessed data splits, and generated samples.

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

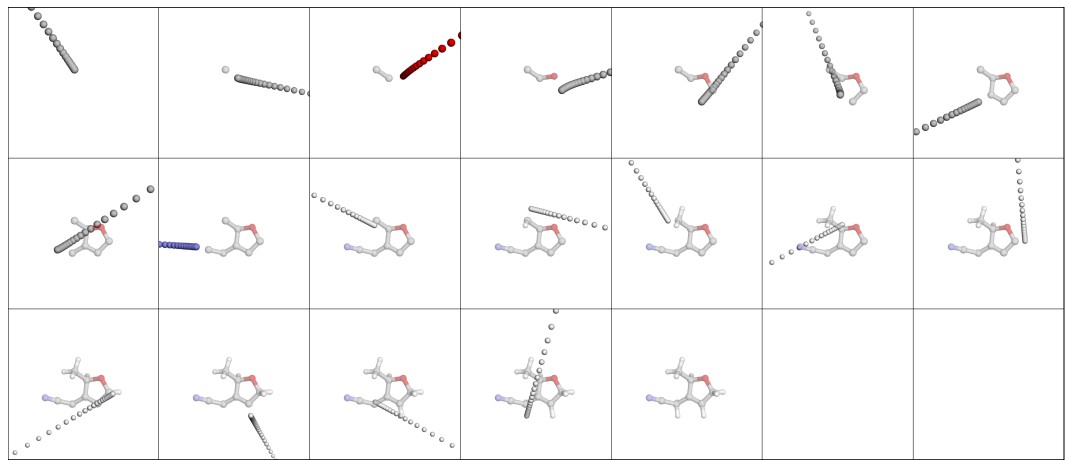

Figure 5: QUETZAL generates 3D molecules by iteratively predicting the next atom's discrete type and continuous 3D position. The continuous trajectories of the DiffMLP are shown at every step.

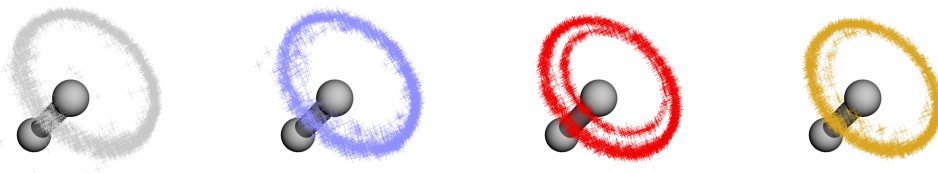

Figure 6: Next position distributions for carbon, nitrogen, oxygen, and fluorine. The model learns to make symmetric predictions from data augmentation.

## A KARRAS ET AL. (2022) DIFFUSION FRAMEWORK

Following the framework of Karras et al. (2022), we learn the optimal denoiser $D_\theta$ ($x$-prediction), which equivalently tells us the score $\nabla_x \log p_t(x)$. To noise the input, we sample timesteps from a log-normal distribution $\ln t \sim \mathcal{N}(-1.2, 1.2^2)$ and directly add noise: $x^{\text{noisy}} = x + t\varepsilon$, where $\varepsilon \sim \mathcal{N}(0, I)$. The neural network is preconditioned using the following reparameterization,

$$D_\theta(t, x^{\text{noisy}}) = \frac{\sigma_{\text{data}}^2}{t^2 + \sigma_{\text{data}}^2} x^{\text{noisy}} + \frac{t\sigma_{\text{data}}}{\sqrt{t^2 + \sigma_{\text{data}}^2}} F_\theta\left(\frac{x^{\text{noisy}}}{\sqrt{t^2 + \sigma_{\text{data}}^2}}, \frac{1}{4}\ln t\right), \qquad (12)$$

where $\sigma_{\text{data}}$ is a hyperparameter set to the standard deviation of the coordinates depending on the dataset, and $F_\theta$ is the actual DiffMLP. The denoising score matching loss of Equation (9) on each timestep is weighted by $(t^2 + \sigma_{\text{data}}^2)/(t\sigma_{\text{data}})^2$.

For sampling, we start with a random sample $x \sim \mathcal{N}(0, \sigma_{\text{max}}^2 I)$ and deterministically integrate Equation (5) using Heun's method, which evaluates $D_\theta$ twice per integration timestep for better accuracy. The number of discretized timesteps is significantly reduced by geometrically spacing them:

$$t_i = \left(\sigma_{\text{max}}^{1/\rho} + \frac{i}{N_{\text{diff}} - 1}(\sigma_{\text{min}}^{1/\rho} - \sigma_{\text{max}}^{1/\rho})\right)^\rho, \qquad (13)$$

where $N_{\text{diff}}$ is the number of diffusion steps, $\rho = 7$, $\sigma_{\text{min}} = 10^{-4}$, and $\sigma_{\text{max}} = 80$.

## B EXPERIMENTAL DETAILS

The causal transformer blocks are identical to GPT-2 (Radford et al., 2019), using the implementation of nanoGPT (Karpathy, 2025), except we apply qk-layernorm along the head dimension for

training stability (Chowdhery et al., 2023; Wortsman et al., 2023), and we do not apply a final LayerNorm before output projection. Biases are also disabled in transformer blocks. For an architecture with $L$ transformer blocks, the first $L/2$ blocks are used for predicting $\boldsymbol{h}_i$, while the second $L/2$ blocks are used for predicting $\boldsymbol{z}_i$. To reduce padding, we use sequence packing (Krell et al., 2021).

We use the same train/val/test splits as Hoogeboom et al. (2022), which contain 10,000/17,748/13,083 examples for QM9 and 5,538,014/692,251/692,251 examples for GEOM.

We train models on QM9 for 2000 epochs. We use sequence packing, using the Longest-pack-first histogram-packing algorithm (Krell et al., 2021). Before training, we pack all examples into sequences of size 128, enforcing a maximum of 6 examples per pack. We then batch packs together by concatenating across the length dimension, with a batch size of 180 packs. This procedure of batching packs was necessary for keeping uniform the number of examples per pack, which was vital for efficient and stable training convergence. We train for 2000 epochs (188k steps) on a single A100 40GB GPU, which took a wall-time of 21 hours. We do document masking using FlexAttention (Dong et al., 2024), preventing the model from attending to other examples in the batched pack.

We do gradient clipping to a norm of 1.0. We train with AdamW (Loshchilov & Hutter, 2017) using a learning rate of $4 \times 10^{-4}$, $\beta_1 = 0.9$, $\beta_2 = 0.95$, and weight decay $= 10^{-5}$. We maintain an exponential moving average of the parameters with decay rate 0.999. We use $\sigma_{\text{data}} = 1.4$ for QM9, and $\sigma_{\text{data}} = 2.5$ for GEOM.

Fourier encodings were important for efficient learning and are used in three different parts of the architecture:

1. For embedding coordinates in the transformer, each element of a vector of size 3 is mapped to 256 Fourier channels with bandwidth $b = 20$, before flattening to size 768.

2. For embedding coordinates in the DiffMLP, each element of each vector of size 3 is mapped to 512 Fourier channels with bandwidth $b = 20$, before flattening to size 1536.

3. For embedding timestep in the DiffMLP, a scalar is mapped to $w$ Fourier channels with bandwidth $b = 1$, where $w$ is the width of the DiffMLP.

We use the magnitude-preserving Fourier encodings proposed by Karras et al. (2024). A scalar $x$ is mapped to a vector of Fourier features via $x \mapsto \sqrt{2} \cos\left(2\pi(bf_i x + \varphi_i)\right)$, where $b$ is the bandwidth, and frequencies $f_i \sim \mathcal{N}(0, 1)$ and phases $\varphi_i \sim \mathcal{U}[0, 1]$ are randomly initialized constants.

For GEOM, we train with 4 A100 40GB GPUs for 201 epochs (734k steps), with a wall-time of 80 hours. We use a learning rate of $2 \times 10^{-4}$ per GPU. We pack all examples into sequences of size 512, enforcing a maximum of 10 examples per pack, and use a batch size of 40 packs per batch.

### B.1 METRICS

Garcia Satorras et al. (2021) introduce several metrics for evaluating the quality of generated 3D molecules. They define a lookup table of allowed bond lengths, with thresholds tuned to maximize the validity of each dataset. A molecule is assigned bonds using this lookup table, and then the valency of each atom is checked. An atom is stable if it has the correct valency. A molecule is stable if all of its atoms are stable. Atom stability is the proportion of generated atoms which are stable. Molecule stability is the proportion of generated molecules which are stable. A molecule is valid if its assigned bonds can be parsed by RDKit without failure. Validity is the proportion of generated examples which are valid. If the molecule can be parsed by RDKit, then it can be turned into a SMILES string. Uniqueness is calculated as the number of unique, generated SMILES strings divided by the number of generated molecules.

We use RDKit's xyz2mol (Kim & Kim, 2015), specifically we use `rdkit==2023.03.3` with `rdDetermineBonds.DetermineBonds(mol, charge=0)`. A molecule is valid if this function passes without error and the resulting molecule can be turned into a SMILES string. We found that this version of RDKit reproduces the results of Daigavane et al. (2023), and determines 99.99% of the QM9 training set to be valid, whereas later versions of RDKit only determines 94.78% to be valid. We estimate the first row of Table 2 by computing metrics for 200k random examples from the training set of GEOM.

918
919
920
921
922
923
924
925
926
927
928
929
930
931
932
933
934
935
936
937
938
939
940
941
942
943
944
945
946
947
948
949
950
951
952
953
954
955
956
957
958
959
960
961
962
963
964
965
966
967
968
969
970
971

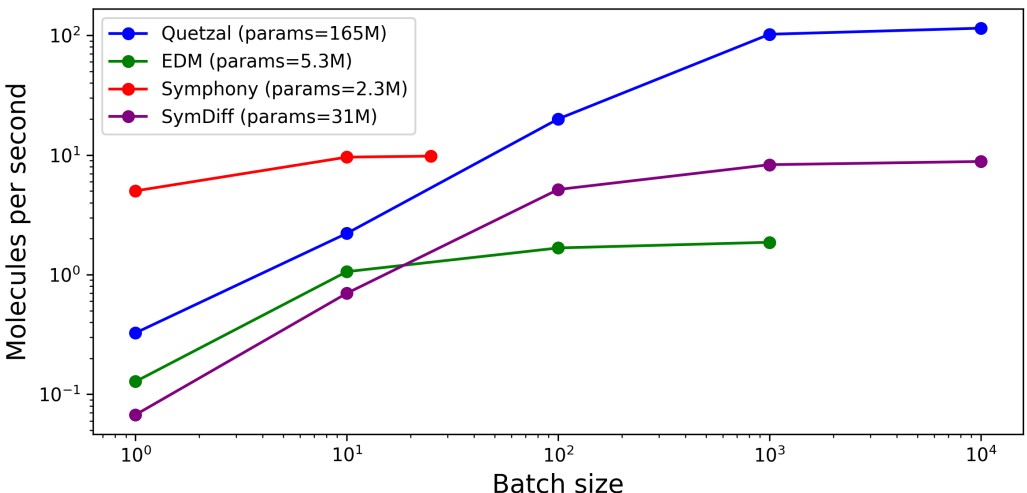

Figure 7: Generation speed of QM9 examples as a function of batch size on a single A100 40GB GPU. Despite having over 5× as many parameters as baselines, QUETZAL scales to large batch sizes at inference time, enabling fast amortized generation.

## B.2 ARCHITECTURE ABLATION

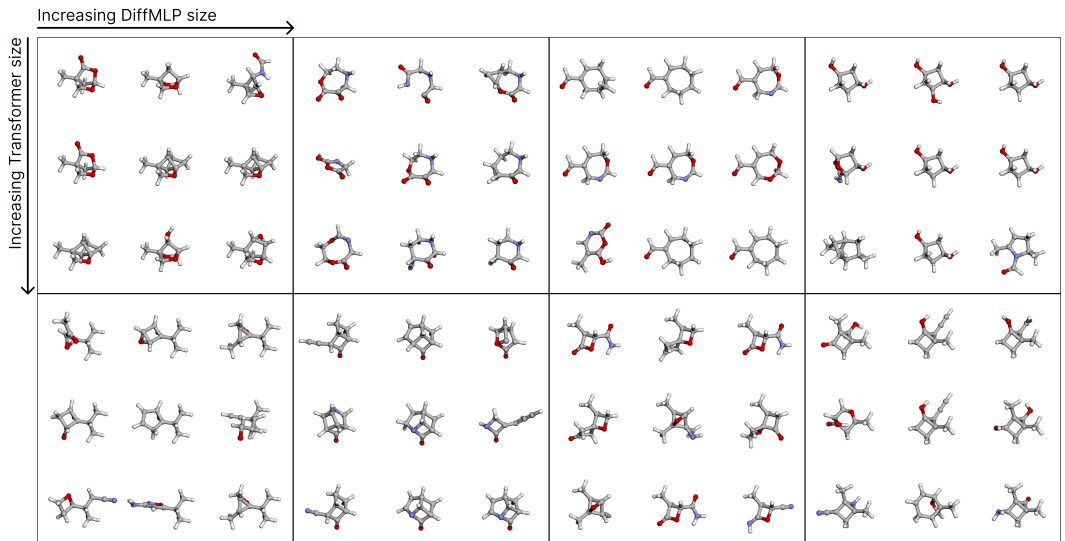

Figure 8: Generated samples using the same random generation seed for different sized models. $N_{\mathrm{diff}} = 30$ diffusion steps are used for each atom. Different models converge to similar molecules. Both the transformer and DiffMLP are important in controlling structure. Model sizes in Table 4.

Table 4: Ablation of transformer and DiffMLP size. $W$ is the transformer width, $H$ is the number of heads, and $L$ is the number of layers. $w$ is the DiffMLP width. Results show mean and standard deviation across 3 evaluation runs. Atom shuffling refers to picking a completely random atom permutation on every training step.

| Transformer size | MLP width | atom stable | mol stable | lookup valid | lookup valid×uniq | xyz2mol valid | xyz2mol valid×uniq |
|---|---|---|---|---|---|---|---|
| $W = 512$ | $w = 512$ | $96.0_{\pm 0.1}$ | $73.8_{\pm 0.4}$ | $87.5_{\pm 0.2}$ | $84.4_{\pm 0.2}$ | $95.8_{\pm 0.1}$ | $91.6_{\pm 0.2}$ |
| $H = 8$ | $w = 1024$ | $97.4_{\pm 0.1}$ | $81.6_{\pm 0.5}$ | $91.6_{\pm 0.2}$ | $88.1_{\pm 0.3}$ | $98.2_{\pm 0.1}$ | $94.0_{\pm 0.3}$ |
| $L = 8$ | $w = 1536$ | $97.7_{\pm 0.0}$ | $83.4_{\pm 0.2}$ | $92.6_{\pm 0.2}$ | $88.9_{\pm 0.1}$ | $98.4_{\pm 0.1}$ | $94.0_{\pm 0.0}$ |
| $W = 640$ | $w = 512$ | $97.1_{\pm 0.1}$ | $80.6_{\pm 0.4}$ | $91.1_{\pm 0.4}$ | $87.3_{\pm 0.5}$ | $96.8_{\pm 0.2}$ | $92.2_{\pm 0.2}$ |
| $H = 10$ | $w = 1024$ | $97.6_{\pm 0.0}$ | $82.9_{\pm 0.3}$ | $92.5_{\pm 0.1}$ | $88.7_{\pm 0.1}$ | $98.4_{\pm 0.1}$ | $93.9_{\pm 0.2}$ |
| $L = 10$ | $w = 1536$ | $98.0_{\pm 0.1}$ | $85.7_{\pm 0.6}$ | $93.8_{\pm 0.4}$ | $89.8_{\pm 0.4}$ | $98.9_{\pm 0.1}$ | $94.0_{\pm 0.2}$ |
| $W = 768$ | $w = 512$ | $96.7_{\pm 0.1}$ | $78.6_{\pm 0.4}$ | $90.3_{\pm 0.1}$ | $85.9_{\pm 0.1}$ | $97.1_{\pm 0.0}$ | $91.7_{\pm 0.2}$ |
| $H = 12$ | $w = 1024$ | $97.9_{\pm 0.0}$ | $85.8_{\pm 0.2}$ | $93.6_{\pm 0.2}$ | $89.3_{\pm 0.2}$ | $98.4_{\pm 0.1}$ | $93.5_{\pm 0.3}$ |
| $L = 12$ | $w = 1536$ | $98.3_{\pm 0.0}$ | $87.6_{\pm 0.3}$ | $94.7_{\pm 0.2}$ | $90.1_{\pm 0.1}$ | $99.1_{\pm 0.0}$ | $94.0_{\pm 0.1}$ |
| with atom shuffling | | $81.2_{\pm 0.1}$ | $12.4_{\pm 0.4}$ | $57.9_{\pm 0.5}$ | $57.6_{\pm 0.6}$ | $81.3_{\pm 0.5}$ | $81.2_{\pm 0.5}$ |
| w/o translations & rotations | | $84.8_{\pm 0.1}$ | $22.0_{\pm 0.1}$ | $48.5_{\pm 0.5}$ | $47.3_{\pm 0.5}$ | $63.0_{\pm 0.3}$ | $60.9_{\pm 0.4}$ |

## B.3 ATOM ORDER ABLATION

We experiment with training QUETZAL on different but local orderings. We reorder atoms in a molecule using stochastic nearest-neighbor traversal:

1. Start from a random atom.

2. Calculate each unvisited atom's minimum distance to any visited atom.

3. Sample the next atom with probability given by a softmax over these distances, with inverse temperature $\beta$.

Table 5: Generation performance of QUETZAL under different atom orders of QM9 only depends on training on sufficiently *local* atom orders. Smaller $\beta$ results in more stochastic traversals; $k$ is the number of traversals cached for training. Results show mean and standard deviation across 3 evaluation runs.

| | atom stable | mol stable | lookup valid | lookup valid×uniq | xyz2mol valid | xyz2mol valid×uniq |
|---|---|---|---|---|---|---|
| $\beta = 0$ (shuffling) | $81.2_{\pm 0.1}$ | $12.4_{\pm 0.4}$ | $57.9_{\pm 0.5}$ | $57.6_{\pm 0.6}$ | $81.3_{\pm 0.5}$ | $81.2_{\pm 0.5}$ |
| $\beta = 1, k = 1$ | $88.9_{\pm 0.2}$ | $45.9_{\pm 0.5}$ | $69.7_{\pm 0.4}$ | $68.3_{\pm 0.2}$ | $83.0_{\pm 0.7}$ | $81.1_{\pm 0.6}$ |
| $\beta = 5, k = 1$ | $95.8_{\pm 0.1}$ | $72.2_{\pm 0.3}$ | $87.4_{\pm 0.2}$ | $84.1_{\pm 0.3}$ | $96.5_{\pm 0.4}$ | $93.0_{\pm 0.3}$ |
| $\beta = 10, k = 1$ | $96.6_{\pm 0.1}$ | $77.0_{\pm 0.4}$ | $89.9_{\pm 0.4}$ | $86.2_{\pm 0.5}$ | $97.5_{\pm 0.0}$ | $93.9_{\pm 0.2}$ |
| $\beta = 10, k = 3$ | $95.4_{\pm 0.1}$ | $69.3_{\pm 0.5}$ | $86.6_{\pm 0.3}$ | $84.3_{\pm 0.3}$ | $95.1_{\pm 0.1}$ | $92.9_{\pm 0.1}$ |
| $\beta = 10, k = 7$ | $95.2_{\pm 0.0}$ | $68.5_{\pm 0.2}$ | $86.2_{\pm 0.4}$ | $84.2_{\pm 0.3}$ | $95.5_{\pm 0.2}$ | $93.7_{\pm 0.2}$ |
| `.xyz` order | $98.7_{\pm 0.0}$ | $90.4_{\pm 0.4}$ | $95.7_{\pm 0.2}$ | $90.2_{\pm 0.2}$ | $98.6_{\pm 0.1}$ | $94.0_{\pm 0.3}$ |

When $\beta = 0$, this algorithm finds a completely random permutation, whereas when $\beta = \infty$, this returns a maximally greedy random permutation. This algorithm produces breadth-first-search (BFS) orders that are qualitatively different from the `.xyz` data ordering, which usually builds the heavy atom backbone in a depth-first-search (DFS) traversal followed by placing all the hydrogen atoms.

Before training, $k$ of these traversals are precomputed for each molecule. We see for $\beta = 10, k = 1$ that although stability and validity metrics shrink, this is compensated by an increase in uniqueness, allowing xyz2mol valid×uniq to match that of the original dataset order. We also observe two trends.

1. As atom orders become less local (as $\beta$ decreases), performance degrades.
2. As more atom orders are seen during training (as $k$ increases), performance slightly degrades.

These trends suggest that showing nonlocal and/or multiple atom orders during training increases the difficulty and diversity of the learning task.

In practice, we find that DFS orders are easier for QUETZAL to learn than BFS orders, since in DFS the next-position distribution will concentrate around the last placed atom, and may be easier to learn and generalize across prefixes. Hence, *we retain the original dataset orders for slightly better performance and for simplicity*. When applying Quetzal to other datasets, we recommend first trying training on the original order, followed by constructing DFS orders.

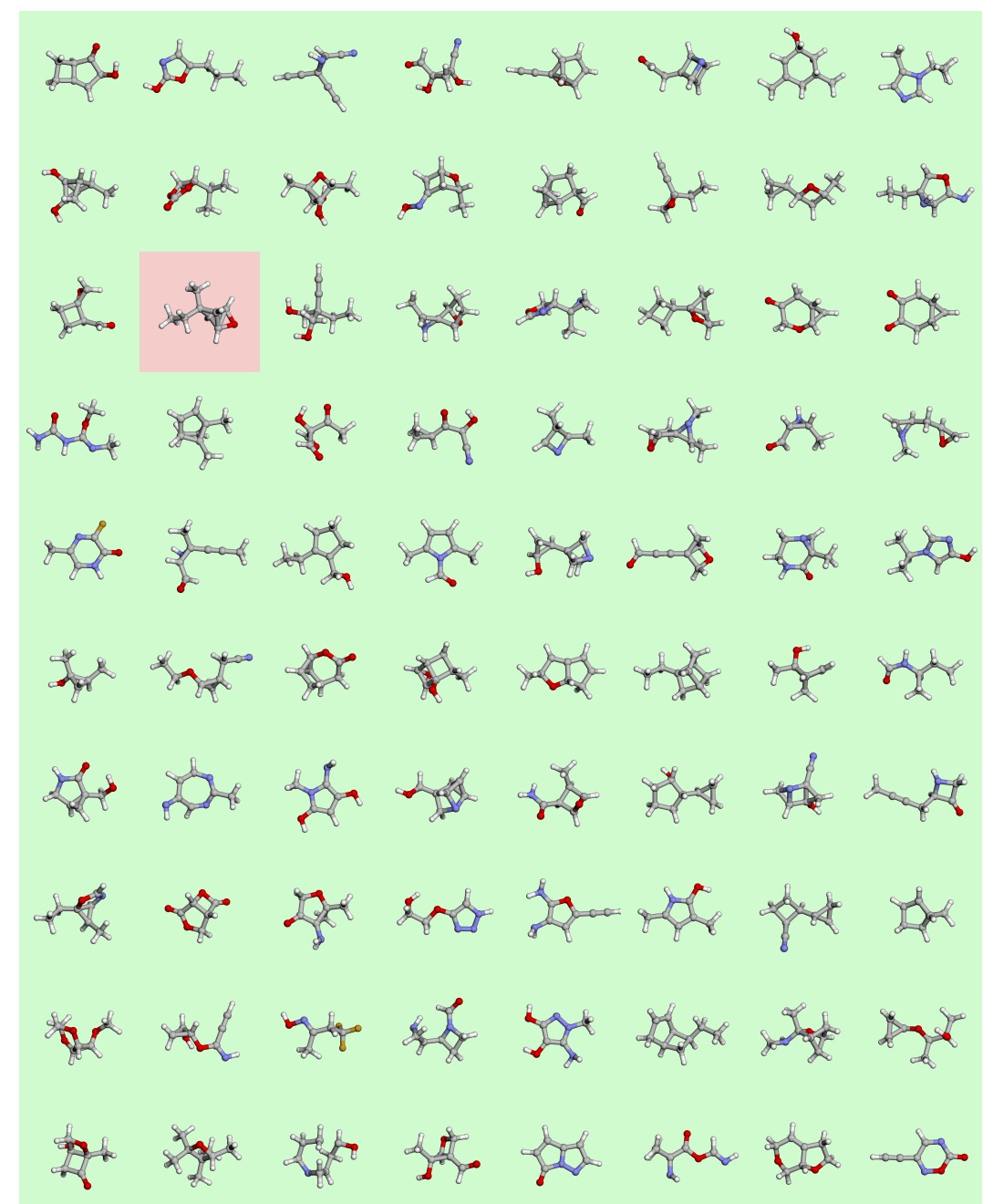

Figure 9: Uncurated generated molecules from QM9. Green/red indicates valid/invalid by xyz2mol.

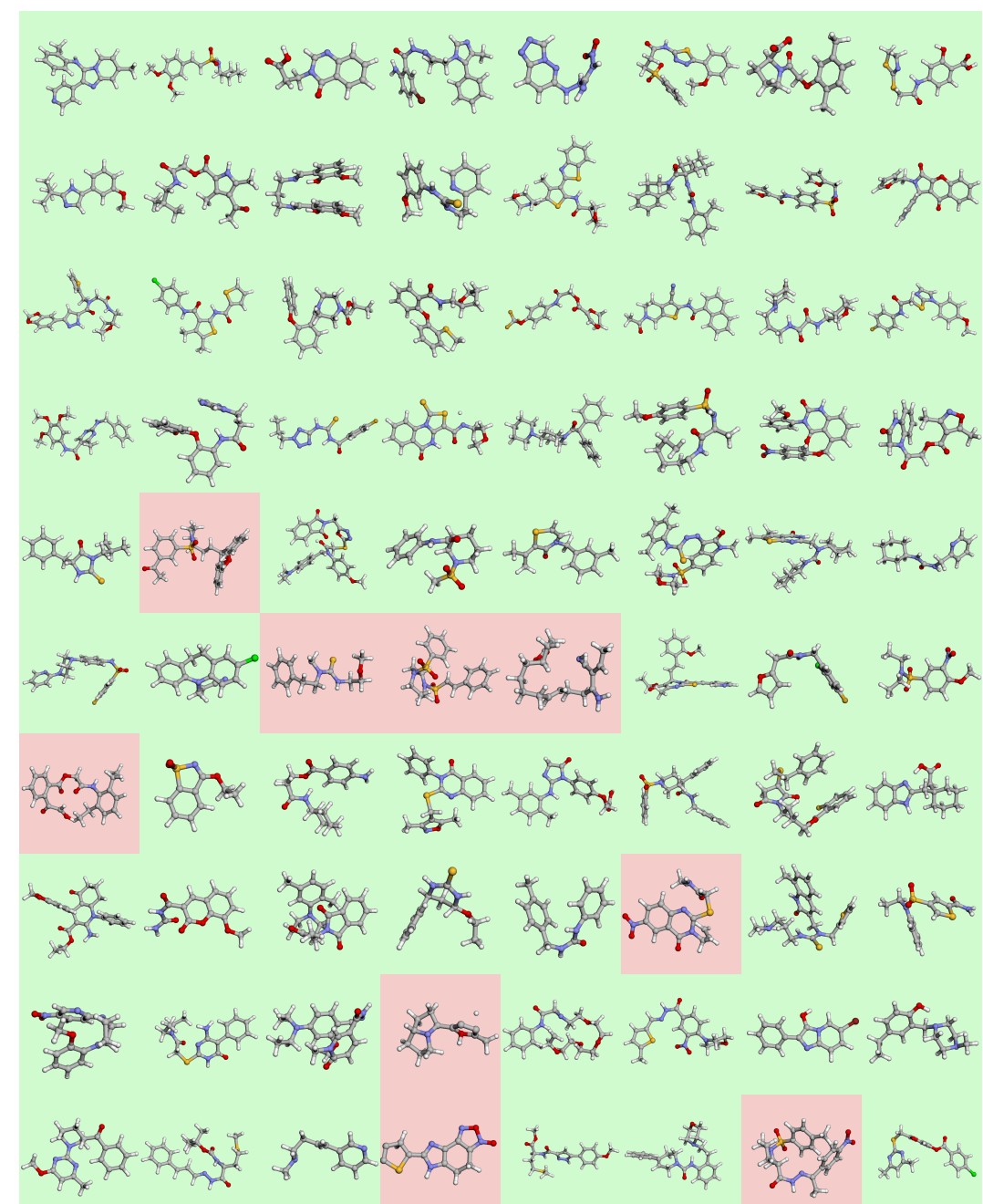

Figure 10: Uncurated generated molecules from GEOM. Green/red indicates valid/invalid by xyz2mol.

## B.4 HYDROGEN DECORATION

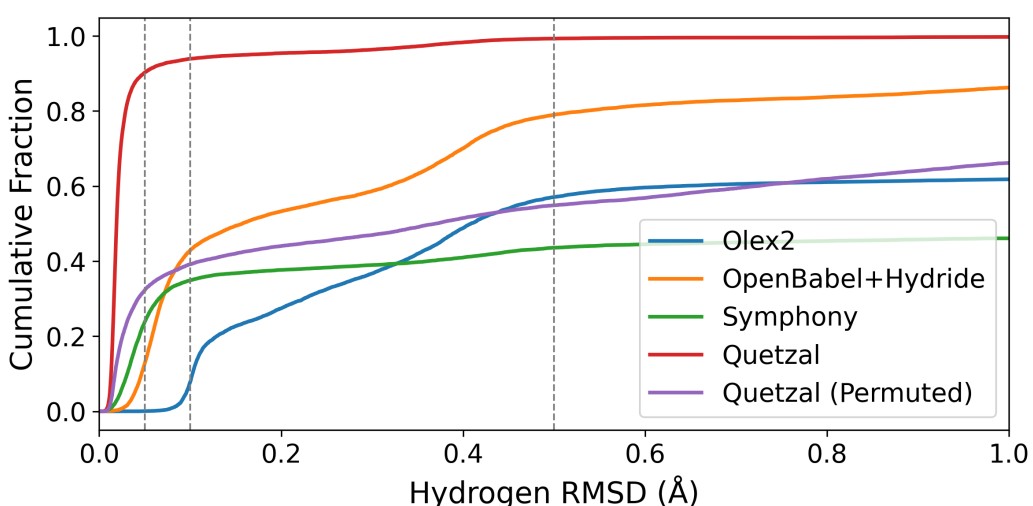

Figure 11: Cumulative distribution functions of RMSD after decorating bare molecules from the test set of QM9. QUETZAL adds hydrogens with very low RMSD for a large majority of the test set. Adding an incorrect number of hydrogens is treated as $\text{RMSD} = \infty$. The vertical dotted lines are the thresholds 0.5, 0.1, 0.05 Å as shown in Table 3. QUETZAL (Permuted) refers to reordering the bare molecule according to a greedy nearest-neighbor traversal. The checkpoint for Symphony appears to be undertrained: https://github.com/atomicarchitects/symphony/blob/3f2c6a7f7983877f4a5f2a0a71328b29bdc553cf/tutorial/workdir/checkpoints/params_best.pkl

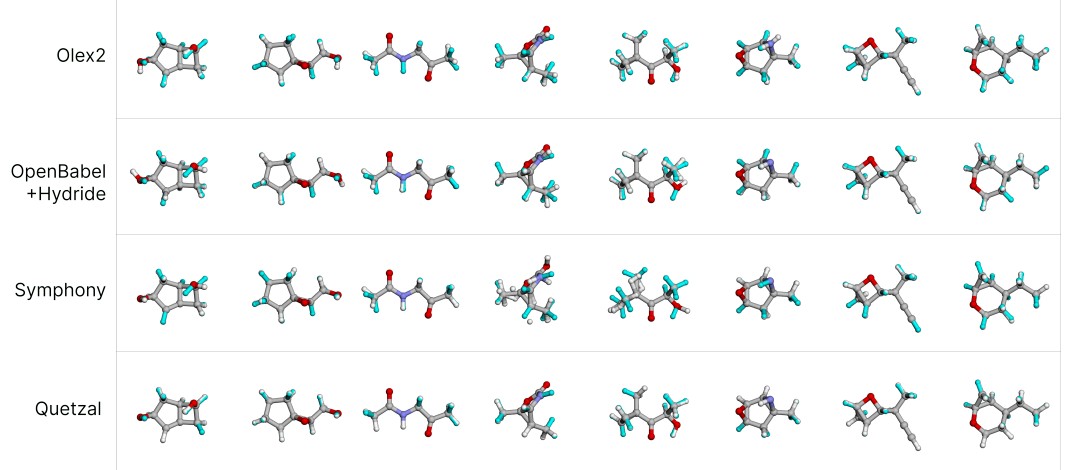

Figure 12: Comparison of methods for adding hydrogens in 3D. The ground truth is displayed in cyan. Accurate hydrogen placement for hydroxyl and methyl groups is difficult.

## B.5 LENGTH GENERALIZATION

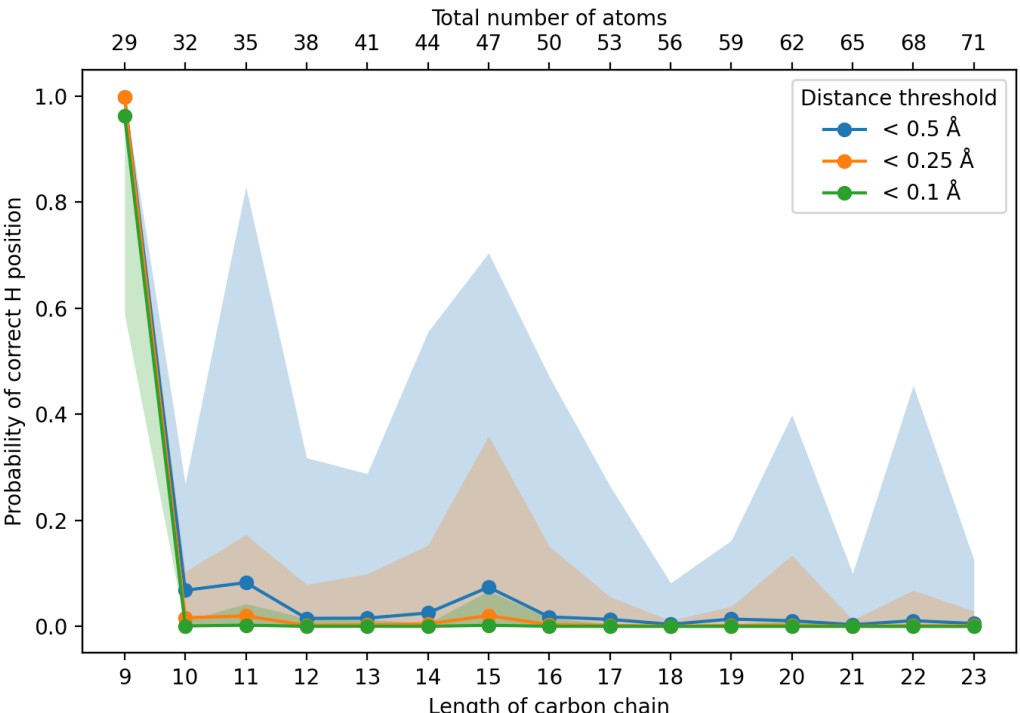

Figure 13: Probability of placing the last hydrogen of a linear alkane chain within RMSD threshold. Probability is measured using 1000 QUETZAL completions for a given RDKit conformer. Mean (solid line) and min/max (shading) are taken over 100 different RDKit conformers.

We test length generalization by forcing QUETZAL to place the last hydrogen on successively longer linear alkanes. We track the percentage of times the sampled hydrogen position is within 0.5, 0.25, and 0.1 Å RMSD of the true hydrogen position.

Beyond nonane, which is in-distribution, the mean probability is low, showing that for most RDKit conformers QUETZAL demonstrates poor but non-zero length generalization. However, the substantial max probability indicates that for some RDKit conformers, QUETZAL is *consistently* able to generalize, overcoming the strong out-of-distribution setting posed by completely untrained positional encodings. These results suggest that Quetzal may generalize to different lengths if combined with techniques such as new positional encodings or context extension methods.

## C    LICENSES

Datasets:

- QM9 (Ramakrishnan et al., 2014): The license status is unclear
- GEOM (Axelrod & Gomez-Bombarelli, 2022): CC0 1.0 Universal

Models:

- EDM (Hoogeboom et al., 2022): MIT License
- Symphony (Daigavane et al., 2023): MIT License
- SymDiff (Zhang et al., 2024): MIT License

