# OpenReview forum: "Scalable Autoregressive 3D Molecule Generation"
_ICLR.cc/2026/Conference — Submitted to ICLR 2026_

### Official Review · Reviewer_p8Z9 · 2025-10-28

**Soundness:** 4
**Presentation:** 4
**Contribution:** 3
**Rating:** 6
**Confidence:** 4

**Summary:**

This paper proposes Quetzal, an autoregressive 3D molecule generative model. The key innovations are modelling the next atom’s position as continuous distribution via a diffusion model, and predicting prefix conditioning vectors with a causal transformer for a cheaper MLP enabling parallel training. Quetzal is benchmarked on QM9 and GEOM-Drugs, showing superior molecule validity with significant speedup.

**Strengths:**

The architecture and overall design makes a lot of sense to me.
For a given atom ordering, Quetzal first computes the prefix conditioning vectors from the current atom positions, using a causal transformer. The prefix conditioning vectors are then fed to predict the next atom types. The use of the causal transformer enables the prediction of the atom types in one forward pass during training, which is more efficient than the existing GNN based methods. Then, the prefix conditioning vectors are fed into a MLP diffusion model, to predict the position of the next atom. The use of the (cheaper) MLP avoids multiple calls to the expensive transformer during diffusion sampling.
The speedups and overall model performance on GEOM-DRUGS and QM9 is quite impressive, showing performance similar to state-of-the-art diffusion models but with significantly faster inference time.

**Weaknesses:**

The reliance on a fixed atom ordering (based on the .xyz file) seems a little worrying (ie Table 4 in Appendix B.2 and the statement “If the bare molecule without hydrogens is reordered, QUETZAL’s performance degrades significantly, as the prefix becomes out-of-distribution.”) The authors are honest about this, which I do commend. But it does suggest a reliance on a canonicalization of the atom order, which is usually not possible in general.

**Questions:**

* In Symphony, they use a nearest-neighbor ordering for the atoms (which is well-defined in the absence of symmetric degeneracy). Perhaps this could help canonicalize your atom ordering instead of using the .xyz file order? I would be interested in seeing such an experiment.
* Symphony also predicts the position distribution in continuous space (without discretization) via spherical harmonics. However, you are correct that for sampling, the distribution is discretized, so I agree with your overall point.
* Some results on length generalization would be good to understand how long the model can go before falling off.

---

> ### Author Response · Authors · 2025-11-24
>
> Thank you for your time and detailed suggestions.
>
> ---
>
> **Canonicalization of atom order and nearest-neighbor ordering**
>
> Thank you for suggesting nearest-neighbor ordering of the atoms. We show in the global response that Quetzal does not rely on training on specific canonical orders; it only relies on sufficiently local orders, which can always be constructed with a nearest-neighbor traversal.
>
> Permutation-invariance (generalizing to different prefix orders for hydrogen decoration and scaffold completion) is a separate concern, and some directions for improvement are given in the response to Reviewer `hs9Q` above.
>
> ---
>
> **Length generalization.**
>
> We test length generalization by forcing Quetzal to place the last hydrogen on successively longer linear alkanes. We track the percentage of times the sampled hydrogen position is within 0.5, 0.25, and 0.1 Å RMSD of the true hydrogen position. See Appendix Figure 13 in the updated paper. We reproduce an except here:
>
> The probability of sampling the correct H position within RMSD threshold is measured by drawing 1000 completions of the same RDKit conformer. The mean and max are taken over 100 different RDKit conformers.
>
> | Num Carbons | Num Atoms | Average Prob < 0.5 Å | Max Prob < 0.5 Å |
> |-------------|-----------|-----------------------|-------------------|
> | 9 | 29 | 0.9984 | 1.0000 |
> | 10 | 32 | 0.0682 | 0.2690 |
> | 11 | 35 | 0.0827 | 0.8290 |
> | 12 | 38 | 0.0153 | 0.3180 |
> | 13 | 41 | 0.0157 | 0.2880 |
>
> Beyond nonane, which is in-distribution, the mean probability is low, showing that for most RDKit conformers Quetzal demonstrates poor but non-zero length generalization. However, the substantial max probability indicates that for some RDKit conformers, Quetzal is *consistently* able to generalize, overcoming the strong out-of-distribution setting posed by completely untrained positional encodings. These results suggest that Quetzal may generalize to different lengths if combined with techniques such as new positional encodings or context extension methods.

---

### Official Review · Reviewer_hs9Q · 2025-10-31

**Soundness:** 3
**Presentation:** 4
**Contribution:** 3
**Rating:** 6
**Confidence:** 4

**Summary:**

This paper introduces QUETZAL, an autoregressive model for 3D molecular generation that predicts atom types and coordinates sequentially using a causal transformer paired with a per-token diffusion MLP. The approach is conceptually simple, highly scalable, and achieves strong sample quality on QM9 and GEOM, with significant generation-speed advantages over diffusion baselines. The authors highlight limitations around atom ordering and rotational/translation augmentation, and they provide candid analysis regarding where their method underperforms (e.g., NLL on QM9) and when comparisons may reflect undertrained models.

This is a technically solid and thoughtfully written paper with good empirical results and strong framing. The permutation-symmetry limitation is significant, but the authors acknowledge it directly. I view this as a worthwhile contribution that advances scalable autoregressive modeling for 3D molecules and provides a useful baseline for future work on order-robust and symmetry-aware variants. Thank you to the authors for the clarity, transparency, and thoroughness.

**Strengths:**

* Clear and well-motivated architecture combining autoregression with continuous coordinate diffusion.
* Competitive or superior sample quality to prior autoregressive methods and close to recent diffusion-based approaches.
* Very fast sampling relative to diffusion models, especially on small molecules.
* Nice demonstrations of variable-size tasks such as hydrogen decoration and scaffold completion without architectural changes.
* Honest and transparent discussion of failure modes and when improvements reflect training scale rather than architecture.
* Thoughtful positioning of autoregressive models as a scalable, flexible alternative rather than a replacement for equivariant diffusion.

**Weaknesses:**

* The core limitation is the lack of permutation symmetry. Performance degrades when atom orderings change, and the model does not generalize to unseen orderings or larger-than-training molecules. This restricts applicability for larger or more diverse chemistries, biomolecules, and materials systems.
* Reliance on fixed xyz ordering and positional inductive bias makes results sensitive to preprocessing and data convention choices.

**Questions:**

1. Did the authors try any strategies to reduce order sensitivity (e.g., randomized orderings during training, curriculum orderings, or masked prefix strategies)? Even negative results would be informative.
2. For hydrogen decoration and scaffold completion, are there systematic diagnostics for how ordering, centering, or orientation perturbations affect performance?
3. Could the authors comment on prospects for integrating partial symmetry handling (e.g., relative positional embeddings or ordering-inference models) while preserving scalability?

---

> ### Author Response · Authors · 2025-11-24
>
> Thank you for your detailed comments, and we appreciate your support of the paper. Your suggestions were helpful in strengthening the discussion section of the paper.
>
> ---
>
> **Performance degrades when atom orderings change.**
>
> We emphasize that the present work focuses on unconditional generation performance, which does not require permutation-invariance, and leave thorough development of prefix-completion tasks for future work.
>
> As discussed in the global response, the performance of Quetzal only depends on finding local orders to train on, which are straightforward to find in any 3D system with nearest-neighbor or DFS traversals. The strong notion of order may even prove to be an advantage in generating linear chains like proteins.
>
> ---
>
> **Generalizing to larger-than-training molecules.**
>
> See the below response to Reviewer `p8Z9` on length generalization.
>
> ---
>
> **Strategies to reduce order sensitivity.**
>
> Thank you for the suggestions. We augment atom orders in the global response.
>
> We did not try curriculum ordering or annealing such as in $\sigma$-GPT (Pannatier et al. 2024) or RAR (Yu et al. 2025), though we believe these to be promising directions for future work.
>
> If by masked-prefix you mean encoder-like autoregression with no attention mask, we did not try this either, since it prevents batching-in-sequence and can be slower to train. However, it may be possible to finetune a causally-pretrained network into an attention-mask-free autoregressive network (Charpentier & Samuel 2024).
>
> Pannatier, A., Courdier, E., & Fleuret, F. (2024, August). σ-gpts: A new approach to autoregressive models. In Joint European Conference on Machine Learning and Knowledge Discovery in Databases (pp. 143-159). Cham: Springer Nature Switzerland.
>
> Yu, Q., He, J., Deng, X., Shen, X., & Chen, L. C. (2025). Randomized autoregressive visual generation. In Proceedings of the IEEE/CVF International Conference on Computer Vision (pp. 18431-18441).
>
> Charpentier, L. G. G., & Samuel, D. (2024). GPT or BERT: why not both?. arXiv preprint arXiv:2410.24159.
>
> ---
>
> **How do ordering, centering, and orientation perturbation affect hydrogen decoration?**
>
> Zero-centering and rototranslations within 3Å do not affect hydrogen decoration performance because of data augmentation. For ordering, we measure Quetzal's performance after permuting the prefix with a nearest-neighbor traversal:
>
> | Method | Correct Num H | % < 0.5Å | % < 0.1Å | % < 0.05Å |
> |---|---|---|---|---|
> | Symphony | 46.9 | 43.6 | 34.9 | 23.8 |
> | Quetzal | 99.8 | 99.3 | 93.9 | 90.2 |
> | Quetzal (Permuted) | 71.2 | 54.9 | 39.2 | 32.3 |
>
> Performance degrades because Quetzal was never trained with BFS orderings, although it still competes with baselines. We haved added this result to Appendix Figure 11.
>
> ---
>
> **How do ordering, centering, and orientation perturbation affect scaffold completion?**
>
> Orientation affects scaffold completion in the following way: if the prefix is located around (0,0,2), then the molecule will tend to build out towards the negative z direction. This is because during training, prefixes that are far from the origin were more likely to have subsequent atoms in the opposite direction. We will add an Appendix figure to qualitatively explain this.
>
> We also generate 1000 completions of benzene with a circular order (123456) and a branched order (321456), and assess validity metrics. The permuted benzene scaffold is strongly out-of-distribution for Quetzal.
>
> | Order | atom stable | lookup valid | lookup valid x uniq | xyz2mol valid | xyz2mol valid x uniq |
> |---|---|---|---|---|---|
> | benzene (123456) | 0.737 | 0.950 | 0.920 | 0.860 | 0.830 |
> | benzene (321456) | 0.670 | 0.329 | 0.327 | 0.133 | 0.133 |
>
> However, we note that Quetzal was not explicitly trained to do hydrogen decoration and scaffold completion. Finetuning with some of the techniques mentioned here would likely improve order robustness.

---

> > ### Author Response · Authors · 2025-11-24
> >
> > **Partial symmetry handling.**
> >
> > We have tried RoPE and observed little difference in unconditional generation results, though we did not test hydrogen decoration, scaffold completion, or length generalization. We believe that standard techniques for improving length generalization of LLMs should also be applicable to Quetzal, such as randomized positional encodings (Ruoss et al. 2023), RoPE context extension methods (Peng et al. 2023), and combining RoPE with qk-normalization (Loshchilov et al. 2024).
> >
> > Ordering-inference methods such as by Wang et al. 2025 may help, but the best atom ordering may ultimately be an empirical result. We also note that, as an autoregressive model, Quetzal could be trained to perform Chain-of-Thought (Wei et al. 2022), whereby the model first reorders the input prefix to a natural order, before finally outputting the completed molecule.
> >
> > Alternatively, one may leverage local permutation symmetries by generating a set of multiple atoms at a time.
> >
> > Wang, Z., Shi, J., Heess, N., Gretton, A., & Titsias, M. K. (2025). Learning-Order Autoregressive Models with Application to Molecular Graph Generation. arXiv preprint arXiv:2503.05979.
> >
> > Ruoss, A., Delétang, G., Genewein, T., Grau-Moya, J., Csordás, R., Bennani, M., ... & Veness, J. (2023). Randomized positional encodings boost length generalization of transformers. arXiv preprint arXiv:2305.16843.
> >
> > Peng, B., Quesnelle, J., Fan, H., & Shippole, E. (2023). Yarn: Efficient context window extension of large language models. arXiv preprint arXiv:2309.00071.
> >
> > Loshchilov, I., Hsieh, C. P., Sun, S., & Ginsburg, B. (2024). ngpt: Normalized transformer with representation learning on the hypersphere. arXiv preprint arXiv:2410.01131.
> >
> > Wei, J., Wang, X., Schuurmans, D., Bosma, M., Xia, F., Chi, E., ... & Zhou, D. (2022). Chain-of-thought prompting elicits reasoning in large language models. Advances in neural information processing systems, 35, 24824-24837.

---

> > > ### Comment · Reviewer_hs9Q · 2025-11-26
> > >
> > > Thank you for your responses to my questions, especially corresponding experiments. My questions have been addressed and I will update my score accordingly.

---

### Official Review · Reviewer_6rAK · 2025-10-31

**Soundness:** 4
**Presentation:** 3
**Contribution:** 2
**Rating:** 4
**Confidence:** 4

**Summary:**

This paper proposes a hybrid approach that combines an autoregressive (AR) model and a diffusion model for generating the 3D structure of molecules. The method leverages a large, computationally intensive AR model to perform a single pass per atom, predicting atom types and associated latent vectors. A small, efficient diffusion model subsequently conditions on these latent vectors to generate the continuous 3D atomic positions.

By integrating the complementary strengths of these two paradigms, the method achieves substantial improvements in sampling efficiency over standard diffusion models, albeit with a larger total parameter count. A key limitation, however, is that the AR component's performance is dependent on the canonical atom ordering found in the source chemical files and is not permutation-invariant.

**Strengths:**

The core idea of hybridizing AR and diffusion models to leverage their respective strengths is compelling. The paper correctly identifies the complementary weaknesses of each paradigm: diffusion models, while powerful, suffer from high sampling costs (modeling all-atom interactions at each denoising step), whereas AR models often struggle to accurately generate continuous 3D coordinates.

The proposed method cleverly capitalizes on the Markovian atom-level decomposition native to AR models. It employs an efficient diffusion model only for the specific task of sampling the 3D position of the current atom, conditioned on a latent vector that aggregates past information. This division of labor is elegant.

The empirical results persuasively demonstrate that this approach achieves competitive generative performance while being substantially faster at sampling than comparable diffusion-based baselines.

**Weaknesses:**

The primary limitation of this work is its dependence on a canonical atom ordering, as found in the chemical data files. The AR component is not permutation-invariant, and its performance degrades significantly when this ordering is broken.

To their credit, the authors are fully transparent about this issue and explicitly report results on permuted data. However, this reliance on a specific data artifact raises significant questions about the method's practical utility and generalizability.

It remains unclear whether this canonical ordering is a common feature of most large-scale chemical databases or a peculiarity of the QM9 and GEOM datasets used for evaluation. While the intricacies of chemical data formats may be outside the typical scope of a core machine learning paper, a method that so fundamentally relies on this data structure warrants a more in-depth discussion of its prevalence and the implications for real-world application.

**Questions:**

1. The paper focuses on the significant gains in sampling efficiency. Could the authors provide a comparison of the training time (e.g., wall-clock time or total FLOPs) against the baselines? This information would provide a more complete picture of the method's overall computational profile.
2. Regarding the "with atom permutations" results reported in Table 4: For the training process, were the atoms permuted once per training instance (i.e., a fixed, new random order) or dynamically in each training epoch (i.e., a different permutation of the same molecule each time it is seen)? This detail is important for understanding the nature of the performance degradation.

---

> ### Author Response · Authors · 2025-11-24
>
> Thank you for the warm commentary and helpful suggestions for the work.
>
> ---
>
> **Reliance on canonical atom ordering.**
>
> In the global response, we show that Quetzal's unconditional generation performance only relies on training on atom orders that are *local* enough -- not on a specific quirk of the `.xyz` order -- and discuss implications for application to other datasets.
>
> ---
>
> **Training time of baselines.**
>
> We compare to SymDiff, since SymDiff reports being faster to train than EDM and GeoLDM. We are unsure if SymDiff chose a batch size that saturates their GPU - if not, then SymDiff’s training speed could be much higher. Batch sizes for Quetzal are approximate, since we pack a variable number of molecules into each sequence.
>
> | Model | Training time (hours) | mol stable | lookup valid x uniq | Hardware | Batch size (molecules) |
> |---|---|---|---|---|---|
> | SymDiff | 64.5 | 89.7 | 94.1 | H100 80GB | 256 |
> | Quetzal (2000 epochs) | 21 | 90.4 | 90.2 | A100 40GB | ~1080 |
> | Quetzal (1000 epochs) | 10.5 | 80.3 | 88.0 | A100 40GB | ~1080 |
>
> These timings suggest that Quetzal offers speedups at both training and sampling time.
>
> ---
>
> **Clarifying atom permutations.**
>
> Atoms were shuffled on every new training batch. We have clarified this in Appendix Table 4.
>
> Performance degradation is expected for training on random-order molecules, since this represents the worst-case. Random orders are highly non-local, jumping from one side of the molecule to the other, and no autoregressive baselines train on random-order. We expect that a permutation-invariant autoregressive model would suffer the same degradation, since any-order generation is in general a harder problem than single-order generation.

---

### Official Review · Reviewer_Dgsk · 2025-11-03

**Soundness:** 2
**Presentation:** 1
**Contribution:** 2
**Rating:** 2
**Confidence:** 4

**Summary:**

The paper proposes QUETZAL, an autoregressive model for 3D molecular generation that builds molecules atom by atom. It combines a causal transformer to predict the next atom’s discrete type with a small diffusion MLP that models its continuous 3D position. This design reduces the number of expensive transformer passes and improves sampling efficiency. QUETZAL achieves generation quality surpassing existing autoregressive baselines and comparable to diffusion models, while also handling variable-size tasks such as hydrogen decoration and scaffold completion without additional training.

**Strengths:**

- Simple and effective combination of autoregression and per-token diffusion.
- Demonstrates strong generation quality and faster inference than diffusion models.
- Naturally supports variable-size molecular generation and related tasks.

**Weaknesses:**

- The method section is not well organized, making it difficult to follow the overall model design and training flow. The paper mixes notation, derivations, and implementation details without clear separation, which obscures the core idea.
- The presentation alternates between describing autoregressive factorization, diffusion objectives, and sampling procedures, but the hierarchy between these components is unclear. It is not obvious at first how the transformer-based atom-type predictor and the DiffMLP position predictor interact or are trained jointly.
- Many implementation notes (e.g., FlashAttention, torch.compile, GPT-2 architecture) appear mid-section and interrupt the theoretical flow. These should be moved to an implementation or appendix section.
- Overall, while the technical components are standard, the lack of structure and guiding intuition in the method section reduces readability and makes it harder to assess novelty and correctness.
- The proposed method appears relatively straightforward, and the paper does not clearly articulate why prior autoregressive approaches were considered unscalable or how QUETZAL specifically overcomes those limitations. The claimed innovation seems incremental without a strong theoretical or architectural justification for the reported improvements.
- Lots of recent papers are not included in the comparison, like GeoBFN (​​https://arxiv.org/abs/2403.15441), which seems to be better on validity metrics

**Questions:**

- Why is the model faster than EDM, since it requires diffusion for each atom, while EDM only requires one diffusion trajectory for sampling
- How is the noising scheme and denoising scheme defined for each atom decoding step

---

> ### Author Response · Authors · 2025-11-24
>
> Thank you for your helpful comments, which we believe have significantly improved the presentation of the work. We appreciate your recognition of the strong empirical results with a simple architecture.
>
> ---
>
> **Method section organization.**
>
> To improve organization and flow, we have added section titles introducing each component of Quetzal in the same order that data is processed by the architecture: prefix embedding, next type prediction, prefix conditioning for diffusion, per-atom diffusion, and combined loss, with a brief digression to introduce diffusion models. **Changes to the PDF are highlighted in blue**.
>
> ---
>
> **How do the transformer and DiffMLP interact?**
>
> The entire network is trained end-to-end, with the per-atom cross-entropy losses added with the per-atom diffusion losses, and summed over the entire sequence. The connection lies in the second transformer stack encoding the prefix into a vector $\mathbf{z}$ to condition the DiffMLP for the next-position distribution.
>
> ---
>
> **Implementation notes interrupt flow.**
>
> We have moved implementation details to the appendix.
>
> ---
>
> **Justification for reported improvements.**
>
> Quetzal makes multiple design decisions to improve over previous autoregressive methods:
> 1. Previous autoregressive methods *discretize* 3D space → Quetzal uses continuous **per-atom diffusion** to predict the next atom’s *continuous* position.
> 2. Previous methods prioritize *architectural equivariance* with poorly-scaling components like tensor products and message-passing → Quetzal uses data augmentation and is therefore free to use deep learning components with *scalable hardware implementations* like FlashAttention.
>
> We now make this clear in the method section on "Inductive biases and scalability".
>
> ---
>
> **Missing baselines.**
>
> We have added GeoBFN to our results on QM9 and GEOM. We emphasize that our main comparison is with *autoregressive models*, where Quetzal advances the state-of-the-art by an absolute 6.8% on lookup valid x unique and 11.5% on xyz2mol valid x unique, beginning to close the gap with all-atom diffusion models. To make this clearer, in our table on QM9 results, we now underline the best metrics out of autoregressive models.
>
> ---
>
> **Reason for speedup over EDM.**
>
> Quetzal generates molecules faster than EDM because of multiple factors:
> 1. Although Quetzal naïvely uses more diffusion steps, each diffusion step is much cheaper since it only requires a small MLP forward pass on an input of size 3, instead of a full transformer or MPNN forward pass on an input of size 3n. Quetzal pays an additional cost of calling a transformer to generate each new atom, but this cost is small for small molecules. The transformer cost increases quadratically as sequence length increases, but can be reduced to linear per-token cost with a kv-cache. We do not use kv-cache in our experiments.
> 2. FlashAttention makes Quetzal’s transformer calls cheaper than EDM’s message-passing calls, while also enabling larger batch sizes than EDM because FlashAttention has a linear cost in memory whereas message-passing has a quadratic cost in memory.
> 3. Quetzal’s diffusion process uses the Elucidating Diffusion Models (Karras et al. 2022) sampling schedule, reducing the number of diffusion steps by ~10x.
>
> We have made this clearer in the results section "Generation efficiency".
>
> ---
>
> **Noising and denoising scheme.**
>
> Diffusion training and sampling closely follows that of Karras et al. 2022, except it is applied per-atom.
>
> During *training*, we sample a timestep $t$ from a log-normal distribution $\ln t\sim \mathcal{N}(-1.2, 1.2^2)$, and directly add noise $x_t = x_0 + t\varepsilon$, where $\varepsilon$ is sampled from a standard normal distribution and $x_0$ is data. The denoiser $D_\theta$ is trained with the loss function $||D_\theta(t, x_t) - x_0||^2$, which determines the score function $\nabla_x \log p_t (x_t) = (D_\theta(t, x_t) - x_t)/t^2$.
>
> To decode an atom’s position, we integrate the ODE $dx = -t\nabla_x \log p_t (x) dt$ with a *deterministic* Heun integrator from $t=80$ to $t=10^{-4}$ with geometrically spaced integration points.
>
> We have made this clearer in Appendix A.
>
> Karras, T., Aittala, M., Aila, T., & Laine, S. (2022). Elucidating the design space of diffusion-based generative models. Advances in neural information processing systems, 35, 26565-26577.

---

> ### Comment · Reviewer_Dgsk · 2025-11-26
>
> I appreciate the author’s response. However, given the current trend in this field towards non-autoregressive methods that demonstrate superior performance and allow for more controllable conditioning, I believe this work may not meet the standards suitable for a poster presentation at the conference.

---

> > ### Author Response · Authors · 2025-11-30
> >
> > We argue that Quetzal's contribution lies in elevating autoregressive models to compete with all-atom diffusion models, while also providing new capabilities that all-atom diffusion approaches do not natively support, such as fast generation, variable-size completion, and exact likelihood estimation. By showing that a simple, scalable autoregressive design can close this quality gap, Quetzal challenges the current trend toward exclusively diffusion-based methods and opens new directions for future research.
> >
> > Moreover, Quetzal's autoregressive structure arguably offers *more controllable conditioning* through natural prefix-conditional generation. Capabilities such as hydrogen decoration and scaffold completion are obtained "for free" and require no architectural modifications, whereas these capabilities are nontrivial to implement with current all-atom diffusion models.

---

### Author Response · Authors · 2025-11-24

We thank all reviewers for their time and thoughtful comments, which we believe have significantly strengthened the work.

Reviewers agree that Quetzal presents strong empirical performance with a “simple and effective” `Dgsk`, “elegant” `6rAK`, and “clear and well-motivated” `hs9Q` design that “makes a lot of sense” `p8Z9`. We are glad that Reviewers `6rAK`, `hs9Q`, and `p8Z9` recognize the transparent and thoughtful presentation of the method in both its strengths and weaknesses.

---

**Reliance on dataset atom orders.**

Reviewers 6rAK, hs9Q, and p8Z9 expressed concern that Quetzal relies on canonical atom orders originating from the dataset, which may not be reliable. We show that **Quetzal only relies on training on atom orders that are *local* enough**, rather than any specific quirk of the original dataset order. Importantly, local atom orders can always be produced by a nearest-neighbor traversal of the molecule.

One method of finding a local ordering is a stochastic nearest-neighbor traversal of the atoms of a molecule:
1. Start from a random atom.
2. Calculate each unvisited atom’s minimum distance to any visited atom.
3. Sample the next atom with probability given by a softmax over these distances, with inverse temperature $\beta$.

When $\beta=0$, this algorithm finds a completely random permutation, whereas when $\beta=\infty$, this returns a maximally greedy random permutation. This algorithm produces breadth-first-search (BFS) orders that are qualitatively different from the .xyz data ordering, which usually builds the heavy atom backbone in a depth-first-search (DFS) traversal followed by placing all the hydrogen atoms.

We experiment with training on stochastic nearest-neighbor traversals for QM9. Before training, $k$ of these traversals are precomputed for each molecule.

| Orders | atom stable | mol stable | lookup valid | lookup valid x uniq | xyz2mol valid | xyz2mol valid x uniq |
|---|---|---|---|---|---|---|
| $\beta=0$ (atom shuffling) | 81.2 | 12.4 | 57.9 | 57.6 | 81.3 | 81.2 |
| $\beta=1, k=1$ | 88.9 | 45.9 | 69.7 | 68.3 | 83.0 | 81.1 |
| $\beta=5, k=1$ | 95.8 | 72.2 | 87.4 | 84.1 | 96.5 | 93 |
| $\beta=10, k=1$ | 96.6 | 77.0 | 89.9 | 86.2 | 97.5 | 93.9 |
| $\beta=10, k=3$ | 95.4 | 69.3 | 86.6 | 84.3 | 95.1 | 92.9 |
| $\beta=10, k=7$ | 95.2 | 68.5 | 86.2 | 84.2 | 95.5 | 93.7 |
| `.xyz` order | 98.7 | 90.4 | 95.7 | 90.2 | 98.6 | 94.0 |

We see for $\beta=10, k=1$ that although stability and validity metrics shrink, this is compensated by an increase in uniqueness, allowing xyz2mol valid x uniq to match that of the original dataset order. We also observe two trends.
1. As atom orders become less local (as $\beta$ decreases), performance degrades.
2. As more atom orders are seen during training (as $k$ increases), performance slightly degrades.

These trends suggest that showing nonlocal and/or multiple atom orders during training increases the difficulty and diversity of the learning task.

In practice, we find that DFS orders are easier for Quetzal to learn than BFS orders, since in DFS the next-position distribution will concentrate around the last placed atom, and may be easier to learn and generalize across prefixes. Hence, *we retain the original dataset orders for slightly better performance and for simplicity*. When applying Quetzal to other datasets, we recommend first trying training on the original order, followed by constructing DFS orders.

---

**Separating generation order and permutation-invariance**

Reviewers `6rAK`, `hs9Q`, and `p8Z9` also expressed concern that Quetzal is not permutation-invariant, which we view as a distinct concern from dependence on generation order because it **only affects performance on prefix-completion tasks, not on unconditional generation**. For example, if asked to perform an infilling task with atoms missing from the middle of the molecule, even a permutation-invariant model may still see this as out-of-distribution if it has never seen infilling prefixes during training.

Generation order is a fundamental challenge with autoregressive models, related to the consistent performance gap between discrete diffusion and autoregressive language models. To be order-independent, a discrete diffusion model must learn to predict the data in every order, which is a harder task than single-order generation.

On the other hand, permutation-invariance *could* be achieved with architecture modifications to positional encodings or finetuning with a non-causal attention mask, but we leave this to future work. Our results on hydrogen decoration and scaffold completion are intended to showcase the capabilities that Quetzal gets "for free" by being an autoregressive model.

---

The paper PDF has been updated, with significant changes highlighted in blue.

---

> ### Comment · Reviewer_hs9Q · 2025-11-26
>
> Thank you for performing the study on performance vs. ordering strategy. This is a very helpful study for zero-ing on the impact of these choices.

---

### Meta-Review · Area_Chair_Djok · 2025-12-19

**Summary:**

The primary lingering concern is the model's inherent lack of permutation invariance, which leaves it fragile to input permutations and dependent on specific "local" atom orderings (like DFS) to function, a constraint that state-of-the-art geometric diffusion models do not share. While the authors demonstrated that consistent ordering strategies can mitigate this, they also confirmed in the rebuttal that the model exhibits poor length generalization, failing to reliably construct molecules larger than those in the training set.

**Reviewer Concerns:**

The authors successfully resolved concerns regarding clarity, efficiency, and ordering sensitivity. In response to Reviewer Dgsk’s criticism of the disorganized method section, the authors restructured the paper, added clear section headers, and moved implementation details to the appendix, which clarified the interaction between the transformer and diffusion components. The authors also substantiated their claims of superior speed by explaining that Quetzal replaces expensive full-graph message passing with lightweight MLP forward passes during diffusion steps, a justification that satisfied the technical queries regarding inference costs.

However, significant concerns remain regarding fundamental invariance, generalization, and methodological relevance. Despite the new traversal experiments, the model inherently lacks permutation invariance. Furthermore, the rebuttal confirmed Reviewer p8Z9's worry about length generalization, with new data showing that the model’s ability to correctly place atoms degrades drastically (from ~90% to ~1.5% accuracy) when generating molecules slightly longer than those in the training set.

**Reviewer Scores:**

Reviewer hs9Q: This reviewer explicitly stated during the rebuttal that the authors had successfully addressed the raised questions regarding order sensitivity and that an updated score was forthcoming. As the paper's strongest proponent the reviewer would likely have raised the score to a definitive acceptance, given that the authors performed the exact ablation study requested (comparing random vs. local ordering) to resolve the primary technical reservation.

Reviewer p8Z9: The primary technical hesitation for this reviewer was the model's reliance on arbitrary .xyz file orderings, leading to a specific suggestion to test a "nearest-neighbor" canonicalization strategy. Since the authors implemented this exact suggestion and demonstrated that the model works effectively with such algorithmic orderings, the concern about "data artifacts" would have been satisfied.

Reviewer 6rAK: This reviewer’s rejection hinged on the belief that the model relied on a "specific data artifact" (the source file order), which was viewed as a significant flaw. Had the reviewer engaged with the rebuttal, they would have seen that the model actually only requires any local traversal (like DFS), meaning the "artifact" issue is solvable by preprocessing.

Reviewer Dgsk: While the authors successfully addressed the complaints about the paper's organization and clearly explained the efficiency advantages over EDM, this reviewer’s objection remained philosophical rather than technical.

---

### Decision · Program_Chairs · 2026-01-26

Reject